# Unveiling the Secret of AdaLN-Zero in Diffusion Transformer

## Abstract

Diffusion transformer (DiT), a rapidly emerging architecture for image generation, has gained much attention. However, despite ongoing efforts to improve its performance, the understanding of DiT remains superficial. In this work, we delve into and investigate a critical conditioning mechanism within DiT, adaLN-Zero, which achieves superior performance compared to adaLN. Our work studies three potential elements driving this performance, including an SE-like structure, zero-initialization, and a "gradual" update order, among which zero-initialization is proved to be the most influential. Building on this insight, we heuristically leverage Gaussian distributions to initialize each condition modulation, termed adaLN-Gaussian, leading to more stable and effective training. Extensive experiments following DiT on ImageNet1K demonstrate the effectiveness and generalization of adaLN-Gaussian, *e.g.*, a notable improvement of 2.16% in FID score over adaLN-Zero.

## 1 Introduction

Diffusion transformer (DiT) (Peebles & Xie, 2023) has recently emerged as a powerful architecture for image synthesis, and has gained vast attention for its superior performance over UNet-based diffusion models (Dhariwal & Nichol, 2021; Rombach et al., 2022). As DiTs continues to drive breakthroughs in image generation, there is a growing interest in pushing its performance boundaries even further. Current efforts could be roughly categorized into two categories: 1) those incorporating advanced techniques (Chu et al., 2024; Ma et al., 2024b; Lu et al., 2024; Tian et al., 2024; Zhu et al., 2024), like VisionLLama (Chu et al., 2024), which introduces language model-based tricks such as RoPE2D (Su et al., 2024) and SwishGLU (Shazeer, 2020), to boost the performance; and 2) those leveraging stronger and more informative conditions (Esser et al., 2024; Chen et al., 2023; 2024a; Ma et al., 2024a; Li et al., 2024), such as PixArt-$\alpha$ (Chen et al., 2023) that extends DiTs to the text-to-image realm to enable more exquisite image generation.

Despite these advances, our understanding of the mechanisms driving DiT's performance remains superficial. One critical aspect that requires further investigation is *adaLN-Zero*, an important conditioning mechanism that significantly enhances DiT's performance compared to the original *adaLN* (20.02 *vs.* 24.13 in FID). Fully understanding the underlying mechanism of adaLN-Zero is essential and may provide deeper insights for further optimizing DiT, especially given the increasing prevalence of DiT in the field of diffusion generation (Karras et al., 2022; Dhariwal & Nichol, 2021; Karras et al., 2024).

In this work, we *uncover the mechanism behind adaLN-Zero's performance boost, providing key insights into DiT's conditioning process.* By studying the differences between adaLN-Zero and adaLN, our analysis studies three elements that collectively contribute to the performance enhancement: 1) an Squeeze-and-Excitation-like (SE-like) structure (Hu et al., 2018), 2) zero-initialized value (a well-optimized location in the optimization space), and 3) a "gradual" update order of model weights. The SE-like structure arises from introducing scaling element $\alpha$ and the latter two stem from adaLN-Zero's zero-initialization strategy for $\alpha$. By empirical experiments, we find that a good zero-initialized location *itself* plays a more significant role among the three elements. We reveal that compared to other initialization, zero-initialization enables the weights that derive $\alpha$ to morphologically more closely approximate the well-trained distribution which resembles a Gaus-

sian distribution. Interestingly, we find all the weights of condition modulations in DiT's blocks gradually form Gaussian-like distributions as training progresses.

Based on these findings, we propose to replace adaLN-Zero by initializing the weights of each condition modulation with Gaussian distributions, which we call adaLN-Gaussian. To validate the effectiveness and generalization of adaLN-Gaussian, we conduct comprehensive experiments following DiT on ImageNet1K (Russakovsky et al., 2015), testing across different training durations, DiT variants, and DiT-based models. Our contributions can be summarized as follows:

• We study three key factors that collectively contribute to the superior performance of adaLN-Zero: an SE-like structure, a good zero-initialized value, and a gradual weight update order. Among them, we find that the a good zero-initialized value plays the most pivotal role.

• Based on the distribution variation of condition modulation weights, we heuristically leverage Gaussian distributions to initialize each condition modulation, termed adaLN-Gaussian.

• Extensive experiments following DiT on ImageNet1K across different settings demonstrate adaLN-Gaussian's effectiveness and generalization, showing a promising pathway for future generative models.

## 2 RELATED WORK

**Transformer in Diffusion.** With the extensively demonstrated scalability and remarkable capabilities of transformers (Vaswani et al., 2017; Dosovitskiy et al., 2020), they have recently been introduced into diffusion generation (Chai et al., 2023; Gao et al., 2023; Mo et al., 2023; Feng et al., 2023; 2024; Bao et al., 2023; Fei et al., 2024; Chen et al., 2024b; Levi et al., 2023; Crowson et al., 2024). Gao et al. propose an asymmetric masking diffusion transformer to explicitly enhance contextual relation learning among object semantic parts. DiffiT (Hatamizadeh et al., 2023) introduces hybrid hierarchical vision transformers with a U-shaped encoder and decoder. More recently, DiT (Peebles & Xie, 2023) replaces the widely-used UNet with transformers in diffusion generation, empirically demonstrating excellent performance and promising scalability. Subsequently, more efforts have been devoted to improving diffusion transformers. Following this research line, FiT (Lu et al., 2024) and VisionLLama (Chu et al., 2024) introduce large language model (LLM) techniques, such as RoPE2D (Su et al., 2024) and SwishGLU, to further enhance DiT. SiT (Ma et al., 2024b) proposes a scalable interpolant framework built on the backbone of DiTs. SD-DiT (Zhu et al., 2024) incorporates masking operations into DiT to accelerate model convergence and improve performance. Pixart-$\alpha$ and Pixart-$\sigma$ (Chen et al., 2023; 2024a) extends DiT to text-to-image synthesis and produces high-quality and exquisite images. U-DiT(Tian et al., 2024) argues that the effectiveness of the U-Net inductive bias is meaningful but has been neglected in DiTs, reintroducing the U-shaped architecture to enhance performance. Different from these efforts, our work is motivated by elevating the understanding of DiT given its great prevalence in the generation realm, and focuses primarily on a crucial conditioning mechanism called adaLN-Zero.

**Weight Initialization.** In a neural network, weight initialization is a crucial operation as it directly determines the initial position in the optimization space (Narkhede et al., 2022). Typically, good initialization aids model training. Common methods include random initialization with (truncated) normal or uniform distributions. Glorot & Bengio introduced a properly scaled uniform distribution for initialization, known as "Xavier" initialization, in Jia et al. (2014). However, this strategy is not suitable for the ReLU activation function (Nair & Hinton, 2010), as ReLU can map negative values to zero, thereby altering the entire variance. To address this, He et al. proposed "Kaiming" initialization, which assumes that half of the neurons are activated while the rest are zero. In the deep learning era, the zero-initialization strategy can be traced back to Goyal et al. (2017), where it was used to accelerate large-scale training potentially via nullifying certain output pathways to implicitly adjust the propagation of backward signals in a supervised learning setting. More recently, it has been widely adopted in diffusion generation (Ho et al., 2020; Rombach et al., 2022) to ease optimization. In DiT (Peebles & Xie, 2023), the impact of zero-initialization is particularly notable, leading to significant performance improvement. Motivated by this, we delve deeper into the underlying reasons, hoping that our findings will inspire further research.

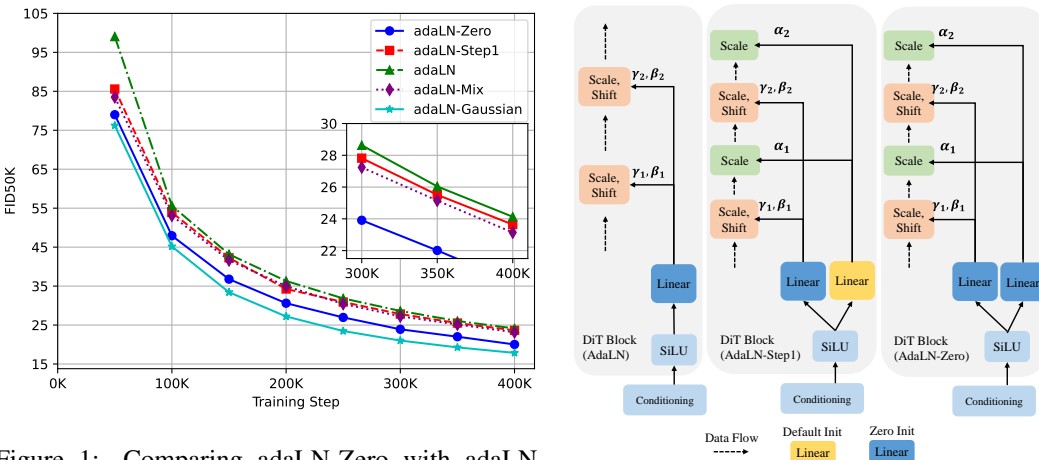

Figure 1: Comparing adaLN-Zero with adaLN, adaLN-Step1, adaLN-Mix, and adaLN-Gaussian on FID50K. We use the largest model DiT-XL/2 in all experiments on ImageNet1K $256 \times 256$.

Figure 2: Illustration of adaLN, adaLN-Step1, and adaLN-Zero. The complete data flow in a DiT block is shown in Alg. 1.

# 3 UNVEILING THE SECRET OF ADALN-ZERO IN DIT

To unveil the underlying mechanism, we perform a detailed comparison between adaLN-Zero and adaLN. In Fig. 2 we find that adaLN-Zero introduces two additional steps: first, it introduces scaling element $\alpha$ (as denoted in DiT) for all transformer blocks; second, it zero-initializes corresponding linear layers to output zero vectors for all $\alpha$. Given these differences, one may naturally wonder: how do these two steps contribute to the performance gap between adaLN-Zero and adaLN in DiT?

## 3.1 DECOUPLING ADALN-ZERO BY EVALUATING STEP ONE IN ISOLATION

To answer this question, we decouple adaLN-Zero by introducing only the first step and initializing the linear layer's weights by default. For convenience, we denote this intermediate state as adaLN-Step1 as shown in Fig. 2 (middle). Then we train the three variants following the same training setting in DiT (Peebles & Xie, 2023) on ImageNet1K for 400K iterations using the largest and best-performing model, *i.e.*, DiT-XL/2. Similarly, we measure FID (Heusel et al., 2017) by using ADM's TensorFlow evaluation suite (Dhariwal & Nichol, 2021) following DiT and compare the performance of adaLN-Step1 with adaLN-Zero and adaLN in Fig. 1. One can see that adaLN-Step1 outperforms adaLN even without zero-initializing the linear layer's weights, indicating that barely introducing scaling element $\alpha$ is beneficial as well. Similar results on Inception Score (IS) (Salimans et al., 2016) could be found in App. A.1. Intuitively, adding scaling element $\alpha$ enhances adaLN's capability of expression, making model optimization easier and more flexible. Upon closer examination from overall structure, module function, and mathematical formula, we speculate that this improvement might be due to a Squeeze-and-Excitation-like (SE-like) architecture (Hu et al., 2018) [1]. Specifically, first, adaLN-Zero and SE module both serve as a side pathway compared to the main path. Second, scaling element $\alpha$ and SE module play a similar role, both of which aim to perform a channel-wise modulation operation. Third, formally, omitting the bias term, we illustrate the formulation of $\alpha$ in DiT in Eq. 1 and SE module (Hu et al., 2018) in Eq. 2, respectively, with slight adjustments to make the two formulas more comparable:

$$F(c) = \underbrace{(c \odot \mathrm{Sigmoid}(c))}_{\mathrm{SiLU}} * W_\alpha = (c \odot (\mathrm{Sigmoid}(I * c))) * W_\alpha, \tag{1}$$

$$SE(c) = (\mathrm{ReLU}(W_1 * c)) * W_2 = (\mathbb{1} \odot (\mathrm{ReLU}(W_1 * c))) * W_2, \tag{2}$$

where $*$ is matrix multiplication, $\odot$ is Hadamard product, and $\mathbb{1}$ is a vector full of element 1. To some extent, it is observed that $F(c)$ shares a similar formulation with $SE(c)$. Given that $SE(c)$ has been extensively demonstrated to enable a general enhancement over various vision tasks (Hu et al., 2018), this similarity may contribute to the improved performance of adaLN-Step1.

On the other hand, it is worth noting that while adaLN-Step1, *i.e.*, the first step, does contribute positively, there remains a large performance disparity between adaLN-Zero and adaLN-Step1. This suggests that the zero-initialization strategy, *i.e.*, the second step, is equally necessary. We explore this further in the next subsection for clarity.

---

[1] In App. A.2, we provide the structure of SE module to better illustrate the similarity.

---

**Algorithm 1** Forward Process of DiT with One Block

---

**Input:** Noise disturbed latent $x$; A simplified DiT: PatchEmbed $W_{pat}$, matrix $W_{\beta_1}$, $W_{\gamma_1}$, and $W_{\alpha_1}$ deriving $\beta_1$, $\gamma_1$, and $\alpha_1$ for the first modulation, a linear layer $W_{att}$ replacing self-attention, matrix $W_{\beta_2}$, $W_{\gamma_2}$, and $W_{\alpha_2}$ deriving $\beta_2$, $\gamma_2$, and $\alpha_2$ for the second modulation, a linear layer $W_{ffm}$ replacing pointwise feedforward, $\gamma_f$, $\beta_f$, and $W_f$ for the modulation and linear layer in FinalLayer, respectively;

**Output:** predicted $\bar{\epsilon}$;

1: $x_p = x * W_{pat}$     # Reshape and patchify $x$
2:
3: # DiT Block
4: $x_{m_1} = x_p \odot (1 + \gamma_1) + \beta_1$     # Modulation
5: $x_{att} = x_{m_1} * W_{att}$     # Replace attention
6: $x_{out_1} = x_{att} \odot \alpha_1 + x_p$     # Skip connection
7: $x_{m_2} = x_{out_1} \odot (1 + \gamma_2) + \beta_2$     # Modulation
8: $x_{ffm} = x_{m_2} * W_{ffm}$     # Replace FFM
9: $x_{out_2} = x_{ffm} \odot \alpha_2 + x_{out_1}$     # Skip connection
10:
11: $x_f = x_{out_2} \odot (1 + \gamma_f) + \beta_f$     # Modulation
12: $\bar{\epsilon} = x_f * W_f$
13: **return** $\bar{\epsilon}$

---

## 3.2 How Zero-initialization Improves the Performance

For a typical initialization strategy, *e.g.*, kaiming initialization (He et al., 2015), its *fundamental* role is to *determine* the initial location of the model in the optimization space. Particularly, in the case of zero-initialization, besides this function, Goyal et al. suggest that it also has an *additional* role. Specifically, it can implicitly adjust the model structure by nullifying certain output pathways at the beginning of training, more importantly, causing the forward/backward signals to initially propagate through the identity shortcut (He et al., 2016), thereby easing the optimization at the start of training (Goyal et al., 2017). However, is this additional role really responsible for the performance gap between adaLN-Zero and adaLN-Step1? To answer this question, we first examine how this additional role specifically impacts optimization through the lens of gradient update. Afterward, we decouple this impact on gradient update during training to highlight the fundamental role of zero-initialization.

### 3.2.1 Zero-initialization's Impact on Gradient Update

Considering the complexity of the DiT model, we make three reliable modifications to simplify our gradient derivation. First, we use only one DiT block, easing the computations of complex chain rules. Second, we replace the multi-head self-attention and pointwise feedforward modules within the DiT block with simple linear transformations, respectively. Though this replacement alters the structure of the DiT block, from the view of backpropagation it does not affect the gradient flow of other modules but itself which is not our emphasis. Therefore this adjustment could be acceptable. Finally, for a linear layer, we omit the bias term in both the forward and backward passes. These alterations significantly simplify our analysis without negatively impacting the conclusions. We formally present the mathematical forward process in Alg. 1. Note that in DiT, LayerNorm is learning-free, so we omit it from our formulation. The process of gradient derivation for each module weight is provided in App. A.3.

To continue our analysis, reviewing the initialization strategy of DiT is necessary. adaLN and adaLN-Zero both initialize the FinalLayer module to zero, indicating that $\gamma_f$ ($W_{\gamma_f}$), $\beta_f$ ($W_{\beta_f}$), and $W_f$ are all zero at the beginning. As shown in Fig. 2, adaLN and adaLN-Zero also zeros out weights of all modulations including $W_{\gamma_1}$, $W_{\beta_1}$, $W_{\gamma_2}$, and $W_{\beta_2}$ in a block, rendering $\gamma_1$, $\beta_1$, $\gamma_2$, and $\beta_2$ zero. A key difference from adaLN is that adaLN-Zero not only introduces $W_{\alpha_1}$ and $W_{\alpha_2}$ to produce scale parameters $\alpha_1$ and $\alpha_2$ (*i.e.*, adaLN-Step1) ***but also zero out $W_{\alpha_1}$ and $W_{\alpha_2}$ to make $\alpha_1$ and $\alpha_2$ become zero.*** See Tab. 1 2nd row.

Therefore, *in this first forward pass*, $W_f = 0$ and output is zero (Eq. 4). Interestingly, *in the first backward pass*, the gradient of $W_f$, *i.e.*, $\frac{\partial \mathcal{L}}{\partial W_f}$, is not zero while the gradients of the rest, *i.e.*, $\frac{\partial \mathcal{L}}{\partial W_{ffm}}$, $\frac{\partial \mathcal{L}}{\partial W_{att}}$, $\frac{\partial \mathcal{L}}{\partial W_{pat}}$, $\frac{\partial \mathcal{L}}{\partial W_{\gamma_f}}$, $\frac{\partial \mathcal{L}}{\partial W_{\beta_f}}$, $\frac{\partial \mathcal{L}}{\partial W_{\alpha_2}}$, $\frac{\partial \mathcal{L}}{\partial W_{\gamma_2}}$, *etc.*, are zero as their gradient formulas all include $W_f$

| Time/Gradient | $\frac{\partial \mathcal{L}}{\partial W_f}$ | $\frac{\partial \mathcal{L}}{\partial W_{ffm}}$ | $\frac{\partial \mathcal{L}}{\partial W_{att}}$ | $\frac{\partial \mathcal{L}}{\partial W_{pat}}$ | $\frac{\partial \mathcal{L}}{\partial W_{\gamma_f}}$ | $\frac{\partial \mathcal{L}}{\partial W_{\beta_f}}$ | $\frac{\partial \mathcal{L}}{\partial W_{\alpha_2}}$ | $\frac{\partial \mathcal{L}}{\partial W_{\gamma_2}}$ | $\frac{\partial \mathcal{L}}{\partial W_{\beta_2}}$ | $\frac{\partial \mathcal{L}}{\partial W_{\alpha_1}}$ | $\frac{\partial \mathcal{L}}{\partial W_{\gamma_1}}$ | $\frac{\partial \mathcal{L}}{\partial W_{\beta_1}}$ |
|---|---|---|---|---|---|---|---|---|---|---|---|---|
| Initial weight | 0 | $W_{ffm}$ | $W_{att}$ | $W_{pat}$ | 0 | 0 | 0 | 0 | 0 | 0 | 0 | 0 |
| 1st iteration | ✓ | ✗ | ✗ | ✗ | ✗ | ✗ | ✗ | ✗ | ✗ | ✗ | ✗ | ✗ |
| 2nd iteration | ✓ | ✗ | ✗ | ✓ | ✓ | ✓ | ✓ | ✗ | ✗ | ✓ | ✗ | ✗ |
| 3rd iteration | ✓ | ✓ | ✓ | ✓ | ✓ | ✓ | ✓ | ✓ | ✓ | ✓ | ✓ | ✓ |

Table 1: Gradient of different weights during training. The first row is the state of parameters' initial weight. 0 means that the weight is zero. ✓ means that the gradient is not zero and the weight effectively updates while ✗ means the gradient is still zero and the weight does not update.

term and $W_f = 0$. Hence, only $W_f$ is updated while the rest weights are kept. So how about the next? *In the second backward pass*, though $W_f$ is not zero, **the zero-initialized $\alpha_1$ and $\alpha_2$ due to adaLN-zero cause** $\frac{\partial \mathcal{L}}{\partial W_{ffm}}$, $\frac{\partial \mathcal{L}}{\partial W_{att}}$, $\frac{\partial \mathcal{L}}{\partial W_{\gamma_2}}$, $\frac{\partial \mathcal{L}}{\partial W_{\beta_2}}$, $\frac{\partial \mathcal{L}}{\partial W_{\gamma_2}}$, **and** $\frac{\partial \mathcal{L}}{\partial W_{\beta_2}}$ **to remain zero.** How about the third iteration? To better illustrate the gradient variation of involved weights, we show the gradient of all weights in the first several iterations in Tab. 1. One can see that all weights do not update together as expected but *gradually* update. Specifically, in the 1st iteration, only $W_f$ updates. ***In the 2nd iteration, only $W_f$, $W_{pat}$, $W_{\gamma_f}$, $W_{\beta_f}$, $W_{\alpha_2}$, and $W_{\alpha_1}$ update, which is what zero-initialization brings to the optimization update.*** In other words, zero-initialization introduces an additional "gradual" update in the initial stage of optimization compared to adaLN-Step1.

***Remark.*** It is worth noting that although our derivation is based on a simplified version of DiT, we corroborate that this update order aligns with that of original DiT variants in which for a typical DiT model (adaLN-Zero), $W_f$ update first, subsequently, $W_{pat}$, $W_{\gamma_f}$, $W_{\beta_f}$, and *all $W_\alpha$* [2] can update, and finally all parameters start to update. This verification demonstrates that our simplification is reasonable and our derivation is right.

### 3.2.2 DECOUPLING THE IMPACT OF ZERO-INITIALIZATION

Based on Sec. 3.2.1, we know that beyond the difference of initial position in the optimization space, the additional distinction between adaLN-Zero and adaLN-Step1 lies in the *second* iteration of gradient optimization, where adaLN-Zero preferentially optimizes $W_f$, $W_{pat}$, $W_{\gamma_f}$, $W_{\beta_f}$, and *all $W_\alpha$* [3] while adaLN-Step1 optimizes all weights. Considering the performance disparity between adaLN-Zero and adaLN-Step1, is this update discrepancy crucial for enhancing model performance? or is it just the zero-initialized position in optimization space that contributes more?

Intuitively, if this update discrepancy is critical, we would see a *significant* performance variation between adaLN-Zero and adaLN-Step1 within the initial few iterations since this discrepancy only occurs during the second iteration [4]. Thus, we evaluate model performance during the early iterations, as shown in Fig. 3. The results indicate minimal performance fluctuation between adaLN-Zero and adaLN-Step1 during the first 160 iterations, suggesting that the discrepancy in update order may not be as critical as initially expected. Similar results can be found for the Inception Score (IS) in App. A.1.

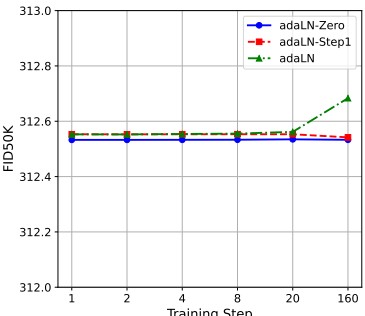

Figure 3: Performance of different variants in the initial stage.

To formally verify our hypothesis, we design an ingenious experiment to decouple the impact of zero-initialization on gradient update. Specifically, considering that the additional effect on the gradient cannot be avoided when zeroing out $W_\alpha$, we adopt the initialization of adaLN-Step1 but enforce the update order of adaLN-Zero simultaneously. We refer to this hybrid strategy as adaLN-Mix and compare its performance with adaLN-Zero and

---

[2]For brevity, we use $W_\alpha$ to denote all $W_{\alpha_1}$ and $W_{\alpha_2}$ in DiT's blocks. $W_\gamma$ and $W_\beta$ are the same.

[3]For adaLN-Step1 (as well as adaLN), the model begins updating all weights after the first iteration, unlike typical initialization strategies where all weights are updated from the very beginning. We will explore the impact of this difference in future work.

[4]In our view, if the discrepancy in gradient updates is crucial, it can *significantly* affect performance in the short term. And as the update period extends, the impact of this discrepancy diminishes.

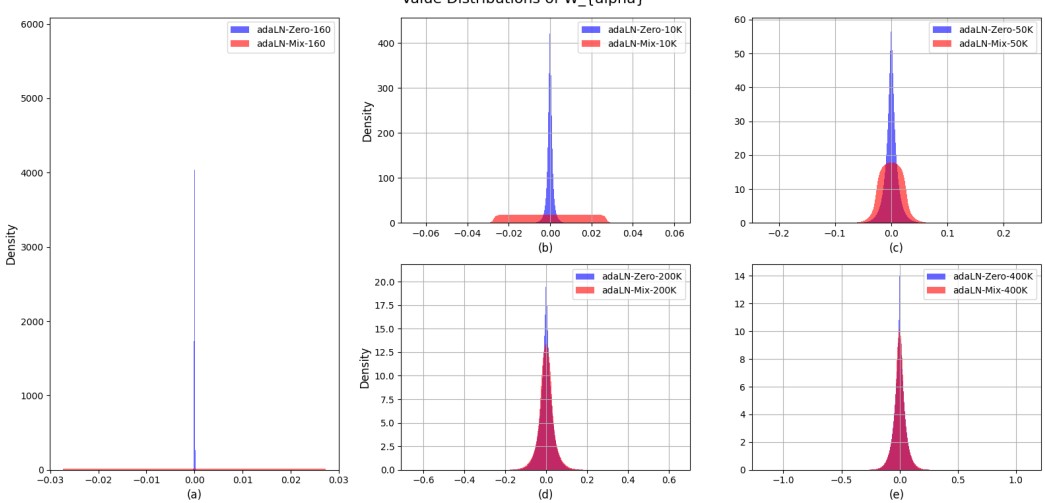

Figure 4: Value distributions of all $W_\alpha$ during the training process. AdaLN-Zero and adaLN-Mix are initialization strategies and 160, 10K, and 50K are timestamps.

adaLN-Step1 in Fig. 1. It is seen that while adaLN-Mix further enhances the performance of adaLN-Step1, it still lags significantly behind adaLN-Zero. This first indicates that the update order resulting from zero-initialization does contribute independently to performance. However, this contribution is not the primary reason for the substantial performance improvement seen in adaLN-Zero. In other words, it is the zero-initialized location that accounts for the remarkable performance difference between adaLN-Zero and adaLN-Mix. Similar results on Inception Score (IS) could be found in App. A.1. Why a zero-initialized location is such important, we put further exploration in the next subsection for clarity.

### 3.3 WHY A ZERO-INITIALIZED LOCATION WINS?

A simple answer might be that zero-initialization avoids introducing noise, as zero is a relatively neutral choice. However, this explanation is neither direct nor fully satisfying, so we aim to unveil a more fundamental reason. Our analysis begins by examining the variation in the weight distribution of all $W_\alpha$ in adaLN-Zero and adaLN-Mix [5], respectively, as training progresses.

As illustrated in Fig. 4, we record the distribution of the entire $W_\alpha$ in 160, 10K, 50K, 200K, and 400K iterations, respectively, to observe the pattern of weight variation over time. At the start, as seen at 160 iterations in Fig. 4 (a), adaLN-Zero exhibits a completely vertical distribution with most values being zero, while adaLN-Mix shows a completely horizontal distribution with a large span of value compared to adaLN-Zero, forming a nearly orthogonal relationship. As the training progresses, (*e.g.*, from 160 to 400K), the distribution of adaLN-Zero remains centered around zero, exhibiting an increasing variance and a concomitant decrease in peak amplitude. Concurrently, the distribution of adaLN-Mix, while expanding peripherally, is also coalescing around zero, culminating in an unimodal structure that is symmetrically centered on zero. Though adaLN-Mix eventually overlaps with the distribution of adaLN-Zero in Fig. 4 (e), the latter's distribution is more compact, with more values concentrated near zero.

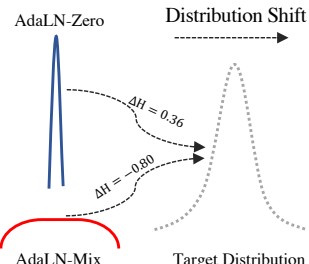

Figure 5: An abstract illustration of the entropy analysis on distribution movement for adaLN-Zero and adaLN-Mix.

Essentially, adaLN-Zero exhibits a more centralized initial parameter distribution, and morphologically, its initial distribution more closely approximates the distribution observed in Fig. 4 (e) than does the adaLN-Mix. This could be the reason why adaLN-Zero converges faster and outperforms adaLN-Mix significantly. From an entropy perspective, our calculations show that when adaLN-Mix is transitioning to the target distribution, *e.g.*, from 10K steps to 50K, entropy decreases by

---

[5]We use adaLN-Mix instead of adaLN-Step1 to eliminate the potential influence of the discrepancy in update order of weights.

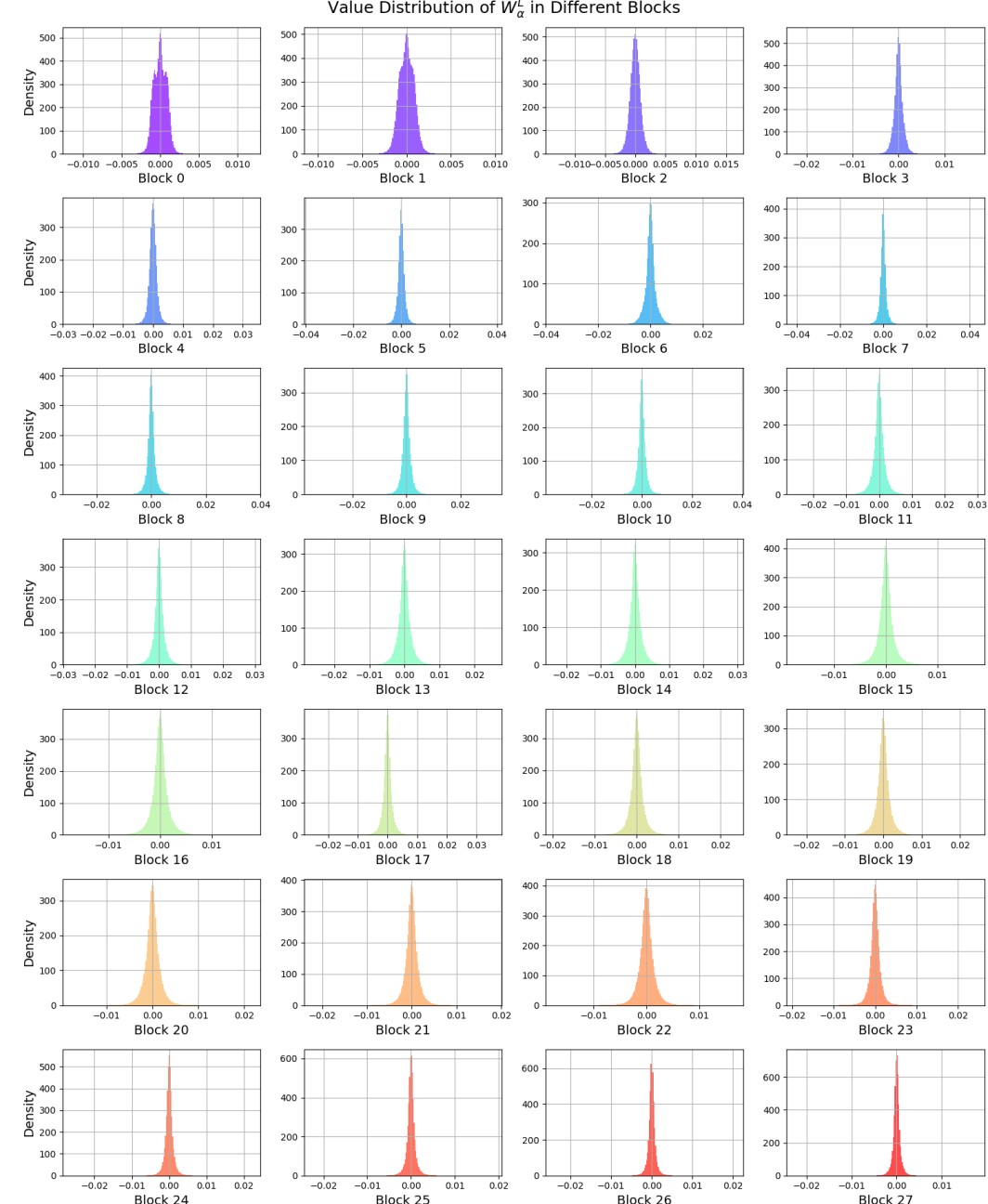

Figure 6: Value distributions of $W_\alpha^L$ in different blocks.

0.8, whereas adaLN-Zero leads to an increase in entropy. Typically, systems tend to evolve towards higher entropy (the second law of thermodynamics). Therefore, adaLN-Zero is comparatively easier to optimize and obtains better performance.

One might question, though we have globally analyzed all $W_\alpha$ in DiT, is it possible that the distribution of $W_\alpha$ across different blocks could differ significantly from the global distribution, considering that zero-initialization is applied on a block-by-block basis? To investigate this, we examine the value distributions of $W_\alpha^L$ ($L$ is block index) of DiT-XL/2 using adaLN-Zero after training for just 10K iterations. As shown in Fig. 6, the distribution of $W_\alpha^L$ in each block closely resembles the pattern observed in Fig. 4 (b), indicating that the functions of $W_\alpha^L$ across different block are likely analogous. This finding also supports the rationale behind *uniformly* zero-initializing $W_\alpha^L$ across different blocks.

***Remark.*** Intuitively speaking, our analysis should have concluded so far. However, we observe that there are other zero-initialized modules in DiT. For the sake of completeness, we provide further

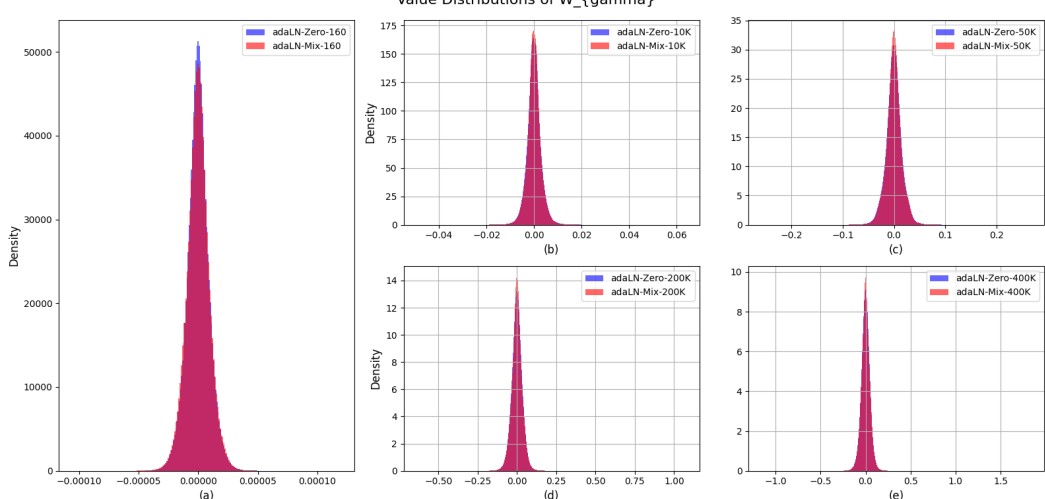

Figure 7: Value distributions of the whole $W_\gamma$ in DiT blocks during the training process.

analysis in the following for clarity. We also show value distributions of other non-zero-initialized DiT modules in App. A.7 and zero convolution in ControlNet (Zhang et al., 2023) in App. A.8.

## 3.4 ANALYSIS ABOUT OTHER ZERO-INITIALIZED MODULES

Recall that in DiT blocks, $W_\gamma$ and $W_\beta$ are zero-initialized in both adaLN-Zero and adaLN-Mix. In addition to that, the FinalLayer module is also zero-initialized at the beginning, indicating that $W_{\gamma_f}$, $W_{\beta_f}$, and $W_f$ are zero in both adaLN-Zero and adaLN-Mix. We want to investigate whether these weights exhibit behavior similar to $W_\alpha$.

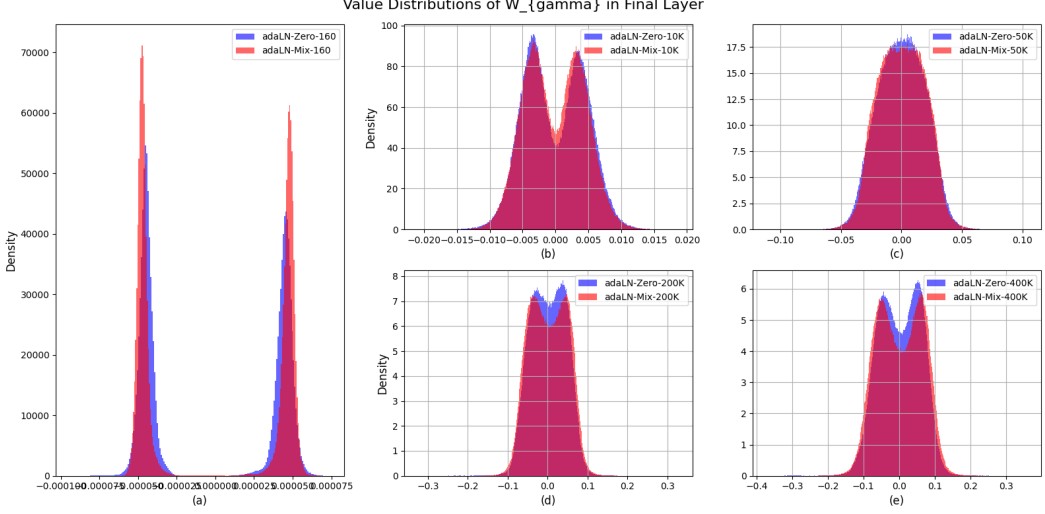

Figure 8: Value distributions of $W_{\gamma_f}$ in FinalLayer during the training process.

***Analysis about $W_\gamma$ and $W_\beta$.*** We present the distribution variations of the entire $W_\gamma$ in DiT blocks as training progresses in Fig. 7. It is observed that, regardless of whether it is adaLN-Zero or adaLN-Mix, $W_\gamma$ rapidly formulates a pattern similar to that of $W_\alpha$ in Fig. 4 at a very early stage. A similar result is observed for $W_\beta$ as detailed in App. A.4. Furthermore, we also show the distribution of $W_\gamma^L$ and $W_\beta^L$ in each DiT block in App. A.5. Basically, the distributions of $W_\gamma^L$ and $W_\beta^L$ in each block share a similar pattern to their global ones as well as that of $W_\alpha$. These results indicate that $W_\gamma^L$ and $W_\beta^L$ may execute analogous functions in DiT blocks.

***Analysis about $W_{\gamma_f}$, $W_{\beta_f}$, and $W_f$.*** As training progresses, we illustrate the variations of value distribution of $W_{\gamma_f}$ in Fig. 8, and that of $W_{\beta_f}$ and $W_f$ in App. A.6. We see that $W_{\gamma_f}$, $W_{\beta_f}$, and $W_f$ exhibit different tendency. For example, $W_{\gamma_f}$ presents a bimodal distribution. These observations suggest that they may not have a consistent update direction compared to $W_\gamma$ and $W_\beta$.

***Remark.*** By comparing the results in Sec. 3.3 and Sec. 3.4, we empirically demonstrate that, although the same zero-initialization strategy is used, weight distributions in different modules may also be discrepant. On the other hand, though weights $W_\alpha$, $W_\gamma$, and $W_\beta$ in the conditioning mechanism are zero-initialized, after a certain number of training steps, they transition from zero distributions to Gaussian-like distributions [6]. This characteristics inspires us to directly initialize these weights with a suitable Gaussian distribution to accelerate training, which we put in the next section to verify.

## 4    ADALN-GAUSSIAN

Our insight is that, as training progresses, the weight distribution gradually transitions from zero to a Gaussian-like distribution. Thus, we can expedite this distribution shift by directly initializing the weights via a Gaussian distribution to potentially accelerate training.

Table 2: Results of different std settings. 0: adaLN-Zero

| Std | FID | IS |
|---|---|---|
| 0 | 78.99 | 14.19 |
| 5e-4 | 80.68 | 13.93 |
| 8e-4 | 79.49 | 14.54 |
| 1e-3 | **76.21** | **15.01** |
| 2e-3 | 78.91 | 14.33 |
| 5e-3 | 79.54 | 14.33 |
| 5e-2 | 84.37 | 13.67 |

To leverage Gaussian distribution to initialize $W_\alpha$, $W_\gamma$, and $W_\beta$, we need to determine the appropriate standard deviation (std), with the mean value defaulting to 0. Intuitively, we can determine the std value by approximating the weight distribution at a specific moment during the training of adaLN-Zero. Moreover, this moment should be neither too late, as initializing $W_\alpha$, $W_\gamma$, and $W_\beta$ at a later stage may impart learned priors incompatible with vanilla weights, nor too early, as there may be minimal difference from zero-initialization (In in App. A.9, we give a detailed result analysis about different std choices in Gaussian initialization.). Therefore, based on Fig. 4, we heuristically select and ablate several std values to uniformly initialize $W_\alpha$, $W_\gamma$, and $W_\beta$ and train each variant for 50K iterations for simplicity. The results are presented in Tab. 2 where $std = 1e-3$ yields the best performance among all variants, verifying the effectiveness of our idea. We denote this initialization method as *adaLN-Gaussian*. The pytorch code below is simple with only one line replaced.

```
for ind, block in enumerate(self.blocks):
    nn.init.constant_(block.adaLN_modulation[-1].bias, 0)
    nn.init.constant_(block.adaLN_modulation[-1].weight,0)
    nn.init.normal_(block.adaLN_modulation[-1].weight, std=0.001)
```

Additionally, we conduct an ablation in Tab. 3 where we apply Gaussian initialization only for $W_\alpha$ [7]. This is the same as adaLN-Step1 but adaLN-Step1 uses default initialization for $W_\alpha$. Hence we denote this variant as adaLN-Step1-Gaussian. Recall that adaLN-Step1 is remarkably inferior to adaLN-Zero while adaLN-Step1-Gaussian here unexpectedly matches and even outperforms adaLN-Zero. This supports our hypothesis that a good initialized position in the optimization space is the key. It also indicates that zero initialization may not be the best choice.

Though the distributions of $W_\alpha$, $W_\gamma$, and $W_\beta$ all resemble Gaussian distribution, in Fig. 4 (b), Fig. 7 (b), and Fig. 11 (b) discrepancies in their shapes persist, *e.g.*, bottom width. Thus, it is more appropriate to select std for each of them independently. We perform a grid search and empirically find that $std(8e-4, 1.2e-3, 8e-4)$ produces the best FID. We denote this initialization as *adaLN-Gaussian-v2* and include the search results of adaLN-Gaussian-v2 in App. A.10 for clarity.

Table 3: Ablation study for $W_\alpha$. 0, 0, 0: adaLN-Zero

| Std ($W_\alpha$, $W_\gamma$, $W_\beta$) | FID | IS |
|---|---|---|
| 0, 0, 0 | 78.99 | 14.19 |
| 1e-3, 0, 0 | 78.62 | 14.42 |
| 1e-3, 1e-3, 1e-3 | 76.21 | 15.01 |

**Longer training time.** To verify the effectiveness of our initialization strategies, as shown in Tab. 4, we train DiT-XL/2 with longer training steps including 400K and 800K on ImageNet1K $256 \times 256$ w/wo CFG. One can see that adaLN-Gaussian outperform adaLN-Zero by a large margin, demon-

---

[6]This similarity may be influenced by the denoising task, which gradually removes Gaussian noise. As our focus is not on the reasons behind these patterns, we leave this exploration as future work.

[7]We observe that $W_\alpha$ plays a critical role in adaLN-Zero compared to adaLN, with its initial value significantly impacting model performance (adaLN-Zero *vs.* adaLN-Step1). Thus, we primarily ablate $W_\alpha$ rather than $W_\gamma$ and $W_\beta$.

| Model | Initialization | CFG | Steps | FID↓ | sFID↓ | IS↑ | Precision↑ | Recall↑ |
|---|---|---|---|---|---|---|---|---|
| *Longer training time*: | | | | | | | | |
| DiT-XL/2 | adaLN-Zero | 1 | 400K | 20.02 | 6.09 | 67.34 | 63.33 | 63.06 |
| DiT-XL/2 | adaLN-Gaussian | 1 | 400K | **17.86** | 6.06 | 73.07 | 64.51 | 62.64 |
| DiT-XL/2 | adaLN-Zero | 1.5 | 400K | 6.15 | 4.60 | 152.70 | 79.92 | 52.28 |
| DiT-XL/2 | adaLN-Gaussian | 1.5 | 400K | **5.28** | 4.62 | 164.62 | 80.75 | 52.65 |
| DiT-XL/2 | adaLN-Zero | 1 | 800K | 14.73 | 6.35 | 86.70 | 65.62 | 63.93 |
| DiT-XL/2 | adaLN-Gaussian | 1 | 800K | **13.14** | 6.11 | 92.98 | 66.50 | 63.92 |
| *Different DiT variants and larger image size*: | | | | | | | | |
| DiT-B/2 | adaLN-Zero | 1 | 400K | 42.72 | 8.29 | 33.28 | 49.02 | 62.80 |
| DiT-B/2 | adaLN-Gaussian | 1 | 400K | **42.55** | 8.13 | 33.82 | 49.05 | 63.30 |
| DiT-L/2 | adaLN-Zero | 1 | 400K | 24.40 | 6.47 | 57.47 | 60.14 | 63.21 |
| DiT-L/2 | adaLN-Gaussian | 1 | 400K | **23.05** | 6.39 | 60.49 | 61.44 | 62.27 |
| DiT-L/4 | adaLN-Zero | 1 | 400K | 45.71 | 9.26 | 32.00 | 46.61 | 60.71 |
| DiT-L/4 | adaLN-Gaussian | 1 | 400K | **44.11** | 9.06 | 33.13 | 47.51 | 61.42 |
| DiT-XL/4$_{512\times512}$ | adaLN-Zero | 1 | 400K | 35.21 | 8.00 | 42.42 | 65.87 | 62.70 |
| DiT-XL/4$_{512\times512}$ | adaLN-Gaussian | 1 | 400K | **34.68** | 7.86 | 42.75 | 65.95 | 61.90 |
| *Different DiT-based models*: | | | | | | | | |
| LlamaVision-XL/2 | adaLN-Zero | 1 | 400K | 21.66 | 6.61 | 65.66 | 60.78 | 63.78 |
| LlamaVision-XL/2 | adaLN-Gaussian | 1 | 400K | **20.26** | 6.20 | 68.82 | 62.06 | 63.90 |
| U-DiT-L | adaLN-Zero | 1 | 200K | 16.28 | 5.50 | 79.91 | 68.31 | 60.44 |
| U-DiT-L | adaLN-Gaussian | 1 | 200K | **15.56** | 5.53 | 82.70 | 68.72 | 60.40 |

Table 4: Comparison on longer training time, different DiT variants, larger image size, and more DiT-based models. We additionally report sFID (Nash et al., 2021) and Precision/Recall (Kynkään-niemi et al., 2019) as secondary metrics following DiT. CFG: Classifier-free guidance. For CFG, we use DiT-XL/2's best guidance value. We use ImageNet1K $256 \times 256$ by default if not specified.

strating the superiority of our initialization strategies. We show more results in Fig. 1. We also show the results of adaLN-Gaussian-v2 in App. A.10.

**Generalization to different DiT variants and larger image size.** To demonstrate the adaLN-Gaussian is a general method, we conduct experiments on several commonly-used DiT variants including DiT-B/2, DiT-L/2, and DiT-L/4. As shown in Tab. 4, we see that adaLN-Gaussian also improves the performance of DiT-B/2, DiT-L/2, and DiT-L/4 though its parameter is set according to DiT-XL/2 and may not be the best setting for these three variants. We further demonstrate the generalization on ImageNet1K $512 \times 512$. These results show the effectiveness of adaLN-Gaussian and imply the great potential of our method after more precise case-by-case adjustments.

**Generalization to other DiT-based models and datasets** [8]. Additionally, we further verify the effectiveness of our method across different DiT-based models. As presented in Tab. 4, it is seen that adaLN-Gaussian is also superior over adaLN-Zero for other DiT-based models including Lla-maVision (Chu et al., 2024) and U-DiT (Tian et al., 2024), demonstrating the generalization of our method. We also show the effectiveness of adaLN-Gaussian on more datasets including Tinyim-agenet (Le & Yang, 2015), AFHQ (Choi et al., 2020), and CelebA-HQ (Karras et al., 2018) and DiT-based SiT (Ma et al., 2024b) in App. A.12.

## 5 CONCLUSION

We study three key factors contributing to the performance discrepancy: an SE-like structure, a good zero-initialized value, and a "gradual" update order of model weights. Moreover, our empirical experiments suggest that a good zero-initialized value *itself* plays a more significant role among these factors. Finally, inspired by the observed distribution variations in condition modulation weights, we propose adaLN-Gaussian which uses Gaussian distributions to initialize condition modulations. We conduct extensive experiments with DiT on ImageNet1K, demonstrating the effectiveness and generalization of adaLN-Gaussian.

---

[8]To save GPU memory, we use the fast version of DiT Github code (https://github.com/chuanyangjin/fast-DiT) featuring gradient checkpointing, mixed precision training, and pre-extracted VAE features, all of which are employed in experiments of Tab. 4. Consequently, though we follow all the training settings, the reported results may be slightly different from that of the original paper.

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

# A   APPENDIX

## A.1   COMPARISON ON INCEPTION SCORE

We also show the comparison on Inception Score (IS) in Fig. 9. We see that adaLN-Step1 outperforms adaLN but is inferior to adaLN-Zero in Fig. 9 (a), indicating again that adding scaling element $\alpha$ is effective in improving model performance. Also, we observe that adaLN-Mix has a marginal enhancement on adaLN-Step1, implying that the discrepancy in gradient update is not the key reason for the large disparity between adaLN-zero and adaLN-Step1. At the same time, in Fig. 9 (b), in the initial iterations when the discrepancy of gradient update happens, we do not see any significant variation on IS, which also demonstrates that the influence of update discrepancy is not critical.

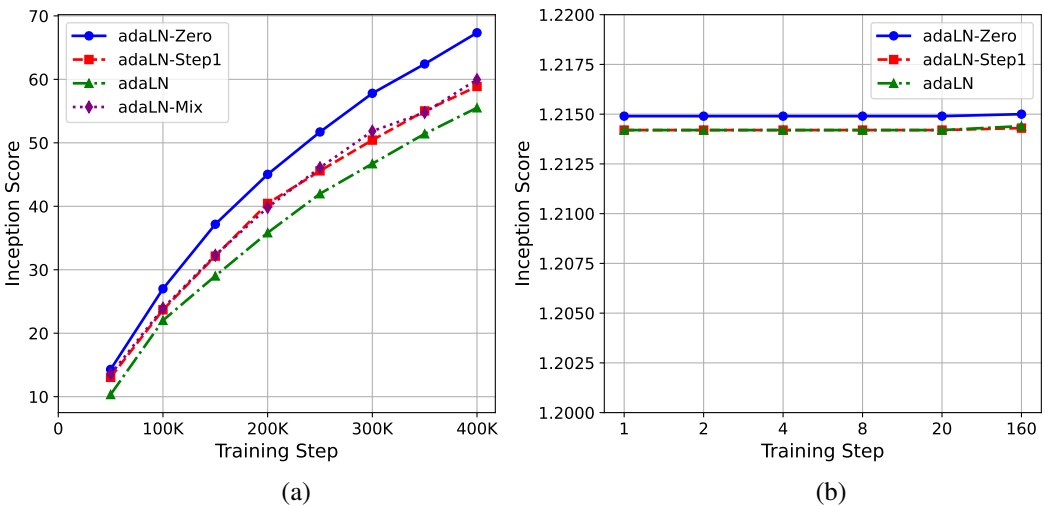

(a)          (b)

Figure 9: Comparing adaLN-Zero with adaLN as well as different initialization strategies on Inception Score (IS). We use the largest model DiT-XL/2 for all the experiments above.

## A.2   THE STRUCTURE OF SQUEEZE-AND-EXCITATION MODULE

In Fig. 10, we illustrate the structure of Squeeze-and-Excitation (SE) module. We can see that SE module serves as a side pathway compared to the main path.

## A.3   GRADIENT DERIVATION OF A SIMPLIFIED DIT

To calculate loss, for simplicity, we only consider MSE loss given the target noise $\epsilon$ sampled from $N(0, I)$ and formulate $\mathcal{L}$ as $\mathcal{L} = \frac{1}{C} \sum_{i=1}^{m} \sum_{j=1}^{n} (\bar{\epsilon}_{ij} - \epsilon_{ij})^2$, where $C = m * n$ and $\epsilon_{ij}$ is the element in row $i$ and column $j$. With this formula, we can obtain $\frac{\partial \mathcal{L}}{\partial \bar{\epsilon}_{ij}} = \frac{2}{C}(\bar{\epsilon}_{ij} - \epsilon_{ij})$. Hence, we deliver a general formula:

Figure 10: The structure of SE module.

$$\frac{\partial \mathcal{L}}{\partial \bar{\epsilon}} = \frac{2}{C}(\bar{\epsilon} - \epsilon). \tag{3}$$

Further, built on Eq. 3, we can also derive the gradient of $W_f$, $W_{ffm}$, $W_{att}$, and $W_{pat}$, $i.e.$, $\frac{\partial \mathcal{L}}{\partial W_f}$, $\frac{\partial \mathcal{L}}{\partial W_{ffm}}$, $\frac{\partial \mathcal{L}}{\partial W_{att}}$, $\frac{\partial \mathcal{L}}{\partial W_{pat}}$, respectively. Before we present

these formulas, we first introduce a substitution to ease our calculation:

$$\bar{\epsilon} = x_f * W_f \tag{4}$$

$$= \{[(x_{m_2} * W_{ffm}) \odot \alpha_2 + x_{out_1}] \odot (1 + \gamma_f) + \beta_f\} * W_f \tag{5}$$

$$= \left\{ \begin{array}{l} [((((x_{m_1} * W_{att}) \odot \alpha_1 + x_p) \odot (1 + \gamma_2) + \beta_2) * W_{ffm}) \odot \alpha_2 \\ + (x_{m_1} * W_{att}) \odot \alpha_1 + x_p] \odot (1 + \gamma_f) + \beta_f \end{array} \right\} * W_f \tag{6}$$

$$= \left\{ \begin{array}{l} [((((((x * W_{pat}) \odot (1 + \gamma_1) + \beta_1) * W_{att}) \odot \alpha_1 + (x * W_{pat})) \\ \odot (1 + \gamma_2) + \beta_2) * W_{ffm}) \odot \alpha_2 + (((x * W_{pat}) \odot (1 + \gamma_1) \\ + \beta_1) * W_{att}) \odot \alpha_1 + (x * W_{pat})] \odot (1 + \gamma_f) + \beta_f \end{array} \right\} * W_f . \tag{7}$$

With the substitution, we can easily derive $\frac{\partial \mathcal{L}}{\partial W_f}$ by using Eq. 4. To derive $\frac{\partial \mathcal{L}}{\partial W_{ffm}}$, we can use Eq. 5. To derive $\frac{\partial \mathcal{L}}{\partial W_{att}}$, we can use Eq. 6. Similarly, to derive $\frac{\partial \mathcal{L}}{\partial W_{pat}}$, we can use Eq. 7. Thus, we calculate the derivation with the help of Laue et al. [9] and present the formula of each below:

$$\frac{\partial \mathcal{L}}{\partial W_f} = x_f^\top * \frac{\partial \mathcal{L}}{\partial \bar{\epsilon}} * I^\top = x_f^\top * \frac{2}{C}(\bar{\epsilon} - \epsilon) , \tag{8}$$

$$\frac{\partial \mathcal{L}}{\partial W_{ffm}} = x_{m_2}^\top * ((\frac{2}{C}(\bar{\epsilon} - \epsilon) * W_f^\top) \odot (1 + \gamma_f) \odot \alpha_2) , \tag{9}$$

$$\frac{\partial \mathcal{L}}{\partial W_{att}} = x_{m_1}^\top \cdot (((T_0 \odot \alpha_2) \cdot W_{ffm}^\top) \odot (1 + \gamma_2) \odot \alpha_1) + x_{m_1}^\top \cdot (T_0 \odot \alpha_1) , \tag{10}$$

where

$$T_0 = (\frac{2}{C}(\bar{\epsilon} - \epsilon) * W_f^\top) \odot (1 + \gamma_f) , \tag{11}$$

and

$$\frac{\partial \mathcal{L}}{\partial W_{pat}} = x^\top \cdot (((T_2 \odot \alpha_1) \cdot W_{att}^\top) \odot (1 + \gamma_1)) + x^\top \cdot T_2 + x^\top \cdot (((T_1 \odot \alpha_1) \cdot W_{att}^\top) \odot (1 + \gamma_1)) + x^\top \cdot T_1 \tag{12}$$

where $T_1$ is

$$T_1 = (\frac{2}{C}(\bar{\epsilon} - \epsilon) * W_f^\top) \odot (1 + \gamma_f) , \tag{13}$$

$T_2$ is

$$T_2 = ((T_1 \odot \alpha_2) * W_{ffm}^\top) \odot (1 + \gamma_2) . \tag{14}$$

Besides these parameters directly involved in input calculations above ($W_f$, $W_{pat}$, $W_{att}$, and $W_{ffm}$), we need to figure out how $\gamma_f$, $\beta_f$, $\gamma_2$, $\beta_2$, $\alpha_2$, $\gamma_1$, $\beta_1$, and $\alpha_1$ update as they also influence the parameters' gradients above as well as the output prediction. Hence, we give their corresponding gradients, respectively (omitting the bias term for simplicity):

$$\frac{\partial \mathcal{L}}{\partial W_{\gamma_f}} = (c \odot \sigma(c))^\top * ((\frac{2}{C}(\bar{\epsilon} - \epsilon) * W_f^\top) \odot x_{out_2}) , \tag{15}$$

$$\frac{\partial \mathcal{L}}{\partial W_{\beta_f}} = (c \odot \sigma(c))^\top * \frac{2}{C}(\bar{\epsilon} - \epsilon) * W_f^\top , \tag{16}$$

$$\frac{\partial \mathcal{L}}{\partial W_{\alpha_2}} = (c \odot \sigma(c))^\top * (T_1 \odot x_{ffm}) , \tag{17}$$

$$\frac{\partial \mathcal{L}}{\partial W_{\gamma_2}} = (c \odot \sigma(c))^\top * (((T_1 \odot \alpha_2) * W_{ffm}^\top) \odot x_{out_1}) , \tag{18}$$

$$\frac{\partial \mathcal{L}}{\partial W_{\beta_2}} = (c \odot \sigma(c))^\top * (T_1 \odot \alpha_2) * W_{ffm}^\top , \tag{19}$$

$$\frac{\partial \mathcal{L}}{\partial W_{\alpha_1}} = (c \odot \sigma(c))^\top * (T_2 \odot T_3) + (c \odot \sigma(c))^\top * (T_1 \odot T_3) , \tag{20}$$

---

[9] https://www.matrixcalculus.org/

Where $T_3$

$$T_3 = \left((\beta_1 + (x * W_{pat}) \odot (1 + \gamma_1)) * W_{att}\right). \tag{21}$$

$$\frac{\partial \mathcal{L}}{\partial W_{\gamma_1}} = (c \odot \sigma(c))^\top * (((T_2 \odot \alpha_1) * W_{att}^\top) \odot x_p) + (c \odot \sigma(c))^\top * (((T_1 \odot \alpha_1) * W_{att}^\top) \odot x_p)), \tag{22}$$

and

$$\frac{\partial \mathcal{L}}{\partial W_{\beta_1}} = (c \odot \sigma(c))^\top * (T_2 \odot \alpha_1) * W_{att}^\top + (c \odot \sigma(c))^\top * (T_1 \odot \alpha_1) * W_{att}^\top, \tag{23}$$

Where $c$ is condition input and $\sigma(\cdot)$ is *sigmoid* function.

## A.4   VALUE DISTRIBUTION OF THE WHOLE $W_\beta$ IN DiT BLOCKS

We present the value distributions of the whole $W_\beta$ of DiT-XL/2 using adaLN-Zero and adaLN-Mix trained for 400K iterations in Fig. 11. Similar to $W_\gamma$, $W_\beta$ quickly formulates a pattern similar to that of $W_\alpha$ in Fig. 4 at a very early stage regardless of whether it is adaLN-Zero or adaLN-Mix.

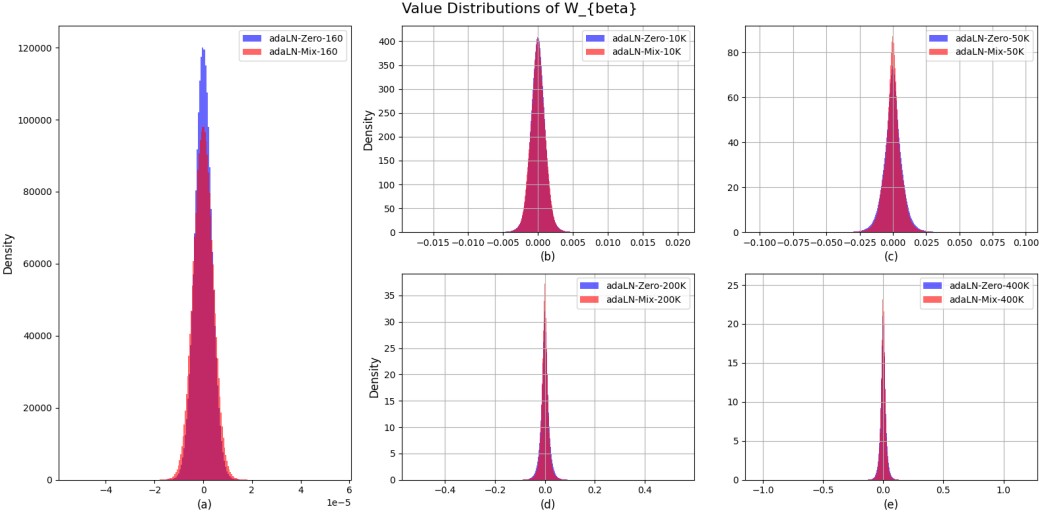

Figure 11: Value distributions of the whole $W_\beta$ in DiT blocks during the training process.

## A.5   VALUE DISTRIBUTIONS OF $W_\gamma^L$ AND $W_\beta^L$ IN DIFFERENT BLOCKS

We also present the value distributions of $W_\gamma^L$ and $W_\beta^L$ in different blocks of DiT-XL/2 using adaLN-Zero trained at a very early stage (for 10K iterations). Fig. 12 and Fig. 13 illustrate the results of $W_\gamma^L$ and $W_\beta^L$, respectively. One can see that, basically, the distributions of $W_\gamma^L$ and $W_\beta^L$ in each block share a similar pattern to their global ones as well as that of $W_\alpha$. Moreover, similar to $W_\alpha^L$, the peak value and bottom width of $W_\gamma^L$ and $W_\beta^L$ vary across blocks and exhibit different std, reflecting the update preference of each block. Built on this observation, this motivates us to initialize $W_\alpha$, $W_\gamma$, and $W_\beta$ with a more sophisticated initialization strategy. More details are in App. A.11.

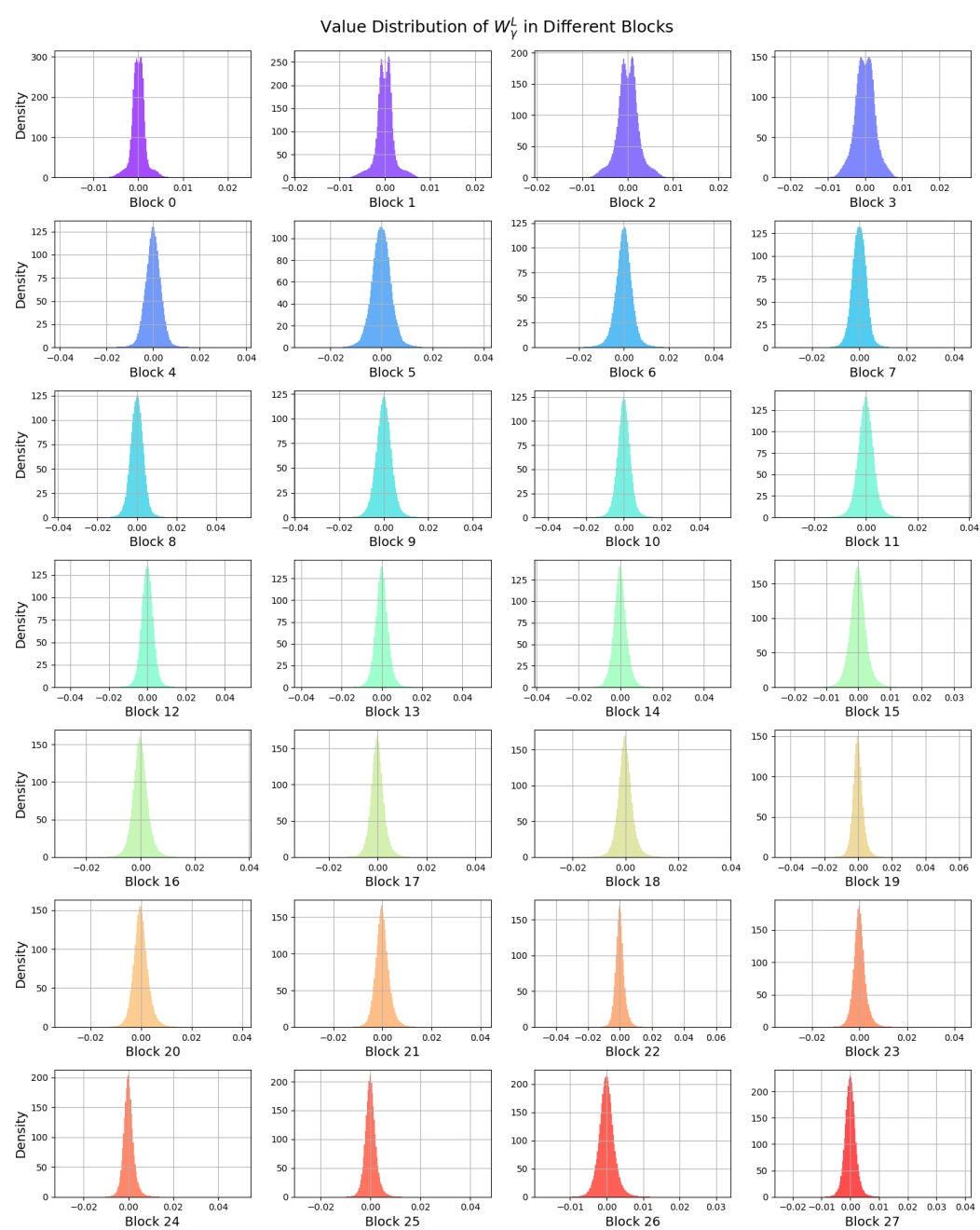

Figure 12: Value distributions of $W_\gamma^L$ in different blocks.

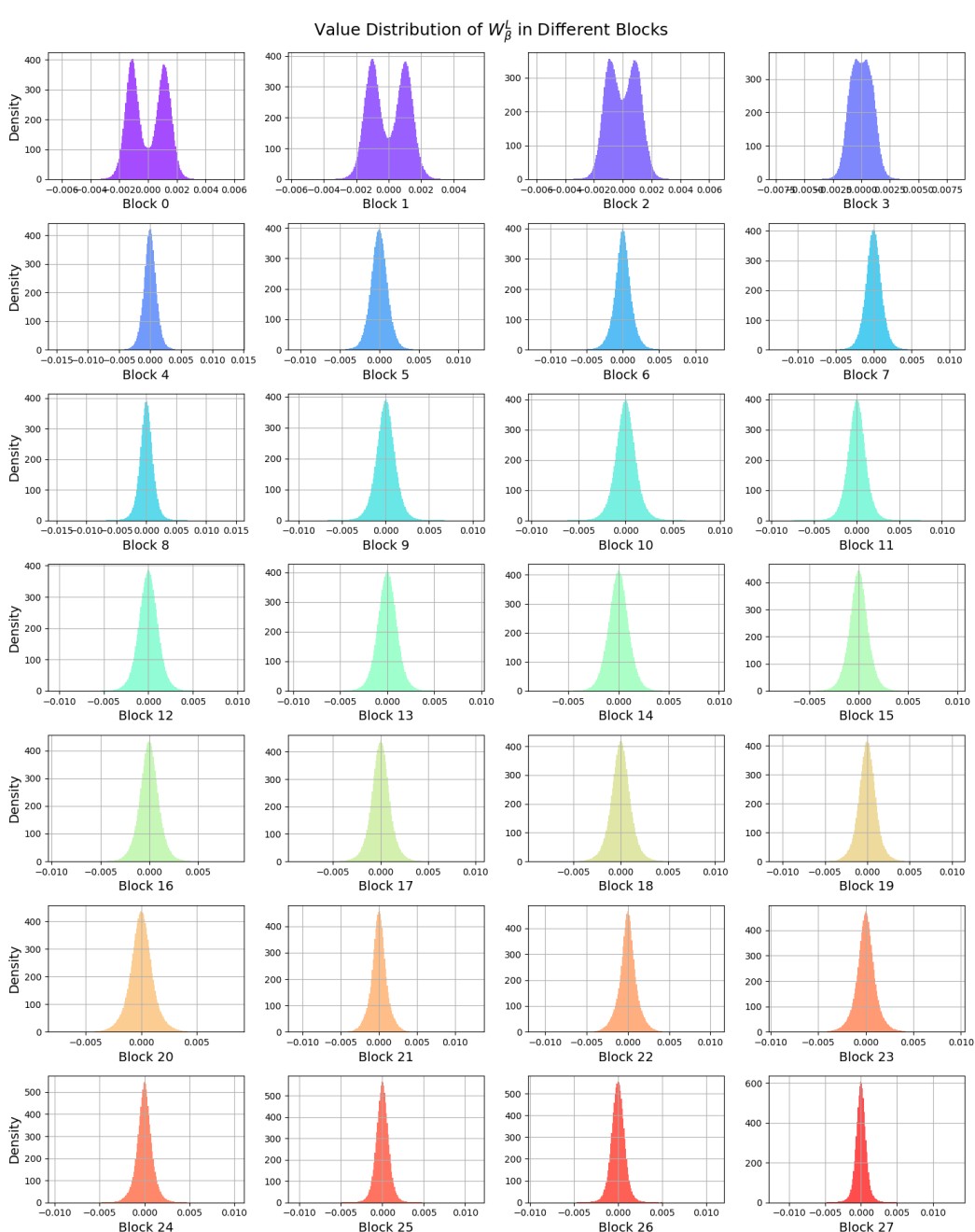

Figure 13: Value distributions of $W_{\beta}^{L}$ in different blocks.

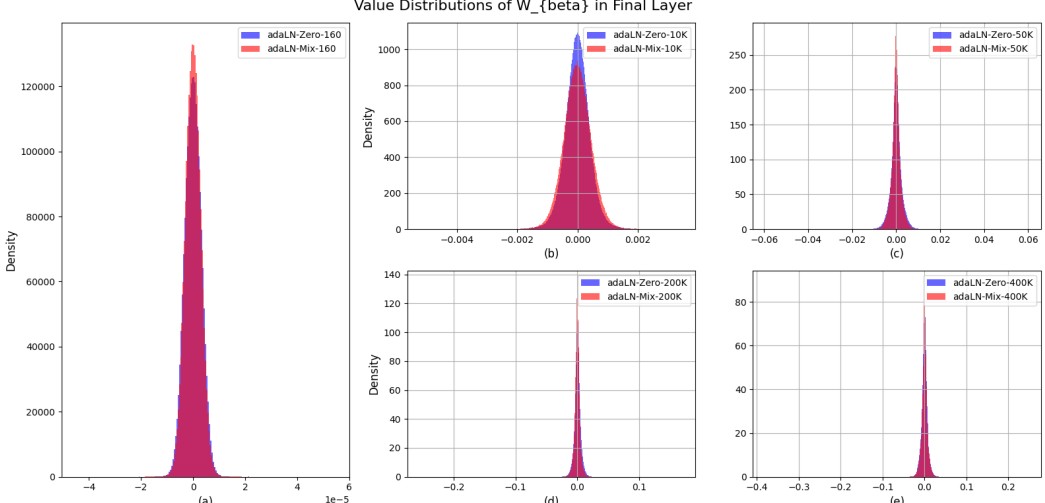

Figure 14: Value distributions of $W_{\beta_f}$ in FinalLayer during the training process.

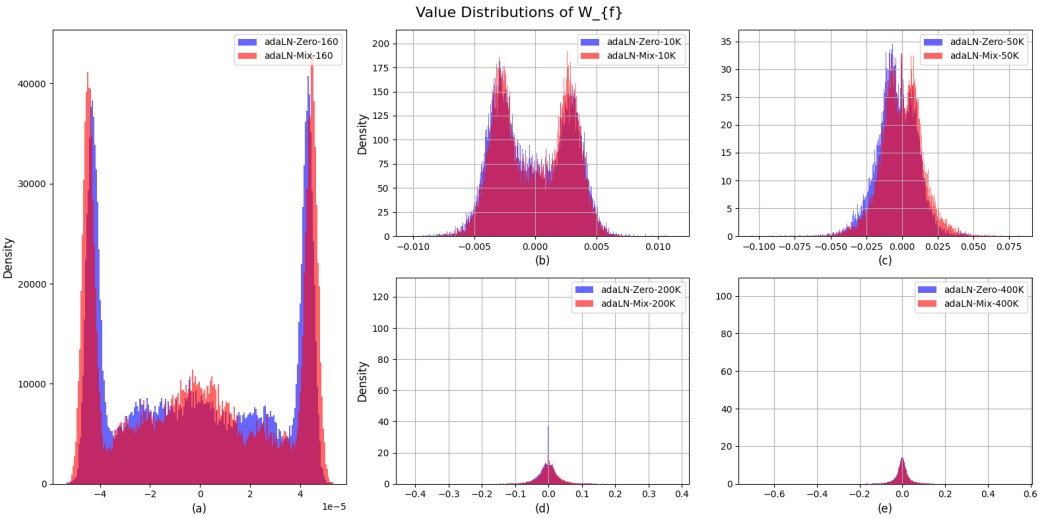

Figure 15: Value distributions of $W_f$ in FinalLayer during the training process.

## A.6 VALUE DISTRIBUTIONS OF $W_{\beta_f}$ AND $W_f$

Fig. 14 and Fig. 15 illustrate the variations of value distribution of $W_{\beta_f}$ and $W_f$ under different training time. We can see that $W_{\beta_f}$ and $W_f$ present completely different variation tendencies. Even though, we also notice that $W_{\beta_f}$ shares a similar pattern to $W_\alpha$ at a very early stage regardless of whether it is adaLN-Zero or adaLN-Mix. This inspires us to explore whether initializing $W_{\beta_f}$ together with $W_\alpha$, $W_\gamma$, and $W_\beta$ could further accelerate training. Based on the setting $std(1e-3, 1e-3, 1e-3)$ for $W_\alpha$, $W_\gamma$, and $W_\beta$, we perform a grid search of std for $W_{\beta_f}$ as shown in Tab. 5. It appears that initializing $W_{\beta_f}$ with a wide range of std values does not enhance the model's performance. In light of these results, we do not consider initializing $W_{\beta_f}$ with Gaussian and keep its original zero-initialization strategy for all the experiments.

Table 5: A grid search of std for $W_{\beta_f}$. 0: AdaLN-Gaussian-v1

| Std | FID | IS |
|---|---|---|
| 0 | 76.21 | 15.01 |
| 2e-4 | 78.22 | 14.53 |
| 5e-4 | 82.05 | 13.78 |
| 1e-3 | 80.43 | 14.03 |
| 2e-3 | 77.45 | 14.74 |
| 3e-3 | 77.09 | 14.84 |
| 4e-3 | 78.47 | 14.39 |

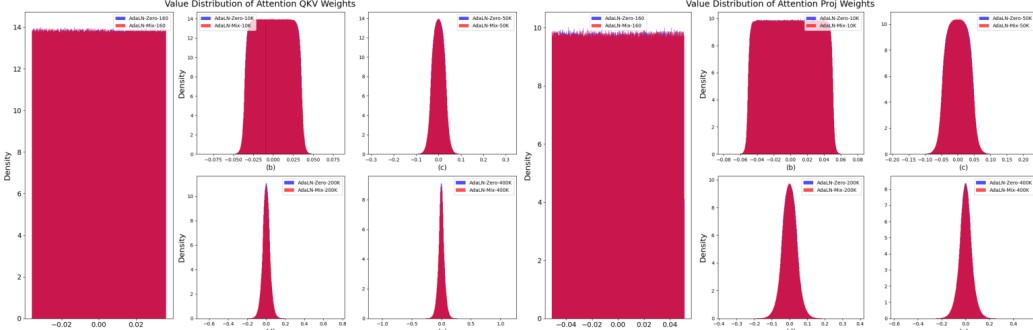

Figure 16: Value distributions of Attention module including qkv and proj during training process.

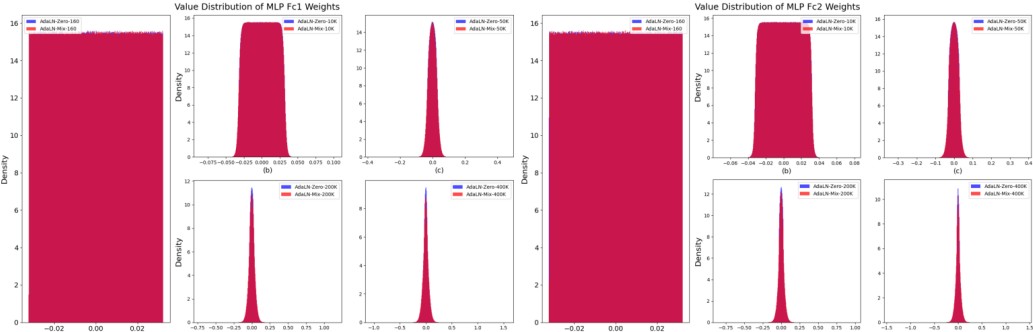

Figure 17: Value distributions of Mlp module including fc1 and fc2 during training process.

## A.7 VALUE DISTRIBUTIONS OF MORE DiT MODULES

We visualize the value distribution of more DiT modules including Attention and Mlp in DiT Block, and PatchEmbed as shown in Fig. 16, Fig. 17, and Fig. 18, respectively. Though they are all initialized with Xavier uniform in DiT, the weight distributions in both Attention and MLP gradually transition to a Gaussian-like distribution while PatchEmbed does not. We also visualize the value distribution of LabelEmbedder and TimestepEmbedder in Fig. 19. We see that after normal initialization done in DiT, their weight distributions consistently show a Gaussian-like distribution. Naturally, we can consider Gaussian initializations for these modules as well except PatchEmbed to accelerate training. For example, we could uniformly use Gaussian initialization for Attention and Mlp in DiT Block. We set the mean to 0 and use several choices for std such as 0.001, 0.01, 0.02, 0.03, and 0.04. We use DiT-XL-2 and train for 50K steps for simplicity. The results are shown in Tab. 6. We see that the performance is inferior to the default initialization. Therefore, more precise hyperparameter tuning may be needed for these modules to further improve the performance in the future.

| Std | Default | 0.001 | 0.01 | 0.02 | 0.03 | 0.04 |
|-----|---------|-------|------|------|------|------|
| FID | 76.21 | 92.09 | 85.28 | 80.89 | 91.21 | 98.50 |

Table 6: Different Gaussian std initialization choices for Attention and Mlp in DiT Block.

## A.8 VALUE DISTRIBUTIONS OF ZERO-CONVOLUTION IN CONTROLNET

Besides adaLN-Zero in DiT, we also consider a similar module in ControlNet (Zhang et al., 2023) called zero convolution. In Fig. 20, we visualize the weight distributions of four widely-used ControlNet variants including Canny, Depth, Pose, and Segmentation. Their distributions are still a Gaussian-like distribution. Hence, is it also beneficial from using Gaussian distribution to initialize these modules in ControlNet? Since it is not our main focus, we leave it as future work.

## A.9 RESULT ANALYSIS ABOUT DIFFERENT STD CHOICES IN GAUSSIAN INITIALIZATION

Intuitively, since the weights of the conditional mechanisms we counted are Gaussian-like distributions, there should exist an optimal std hyperparameter when initializing these weights with Gaussian, and naturally, the values on both sides of this hyperparameter are relatively unsuitable. To

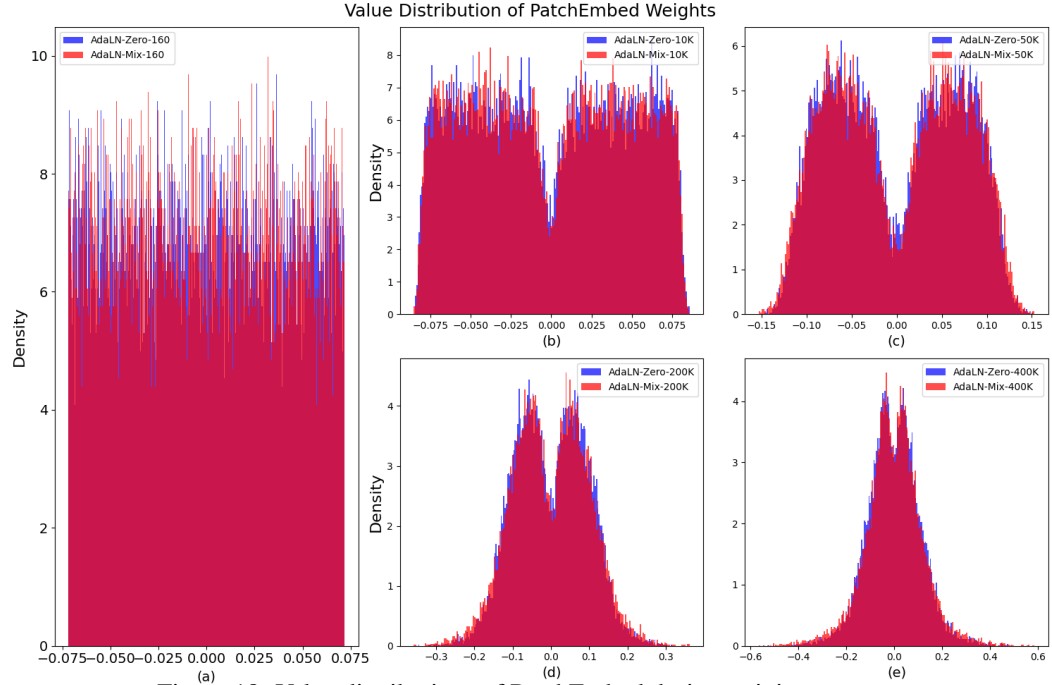

Figure 18: Value distributions of PatchEmbed during training process.

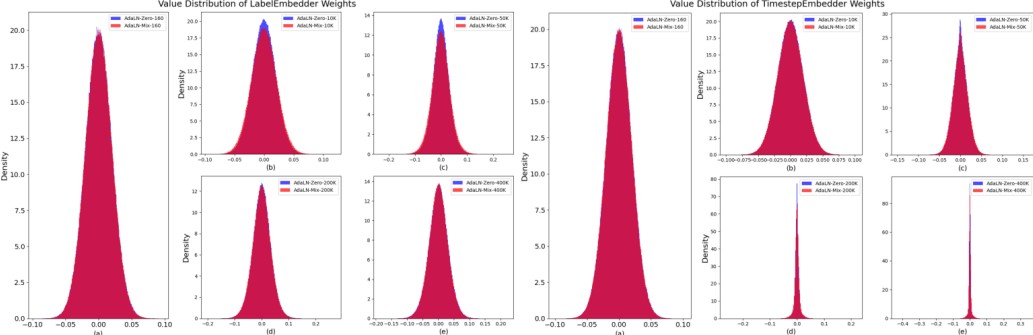

Figure 19: Value distributions of LabelEmbedder and TimestepEmbedder during training process.

some extent, the performance of Gaussian initialization with different std choices in Tab 2 which exhibits a U-shaped trending also proves it. To be more rigorous, we analyze this U-shaped trending by leveraging two representative settings, *i.e.*, $std = 0.0005$ and $std = 0.05$, which the two ends of this U-shaped trending.

We first illustrate their weight distributions of $W_\alpha$ in the conditioning mechanism and compare them with that of adaLN-Zero and adaLN-Gaussian (std=0.001). The results are shown in Fig. 21. We find that a large std $std = 0.05$ presents a relatively uncompact distribution and exhibits a significant discrepancy in distribution shape compared to the rest settings. *This result indicates that a large std may be incompatible with other parameters, resulting in a slow speed of convergence and a poor performance.* Moreover, we consider this a step further. Theoretically, if we further increase the std value, it would become close to the default initialization in adaLN-Step1 (xavier_uniform) while the performance of adaLN-Step1 is also bad.

For a small std std=0.0005, it can be seen that the distribution of $W_\alpha$ is quite similar to that of adaLN-Zero and adaLN-Gaussian ($std = 0.001$). However, there still exists a slight discrepancy. To make this discrepancy clearer, we average the absolute values of the differences between each element in $W_\alpha$ corresponding to $std = 0.0005$ and adaLN-Zero, and $std = 0.0005$ and adaLN-Gaussian. The element-wise averaged results are 0.0121 and 0.0124, respectively. *By comparing the results (0.0121*

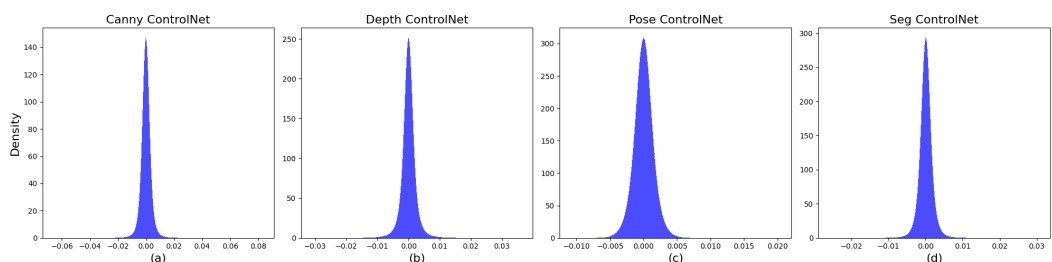

Figure 20: Weight distributions of zero convolution in four ControlNet variants.

*< 0.0124), it is shown that small std leads to weights relatively closer to that of zero-initialization (adaLN-Zero).* And, to some extent, the corresponding performance also proves it where std=0.0005 produces 80.68 for FID, closer to adaLN-Zero (78.99) compared to adaLN-Gaussian (76.21).

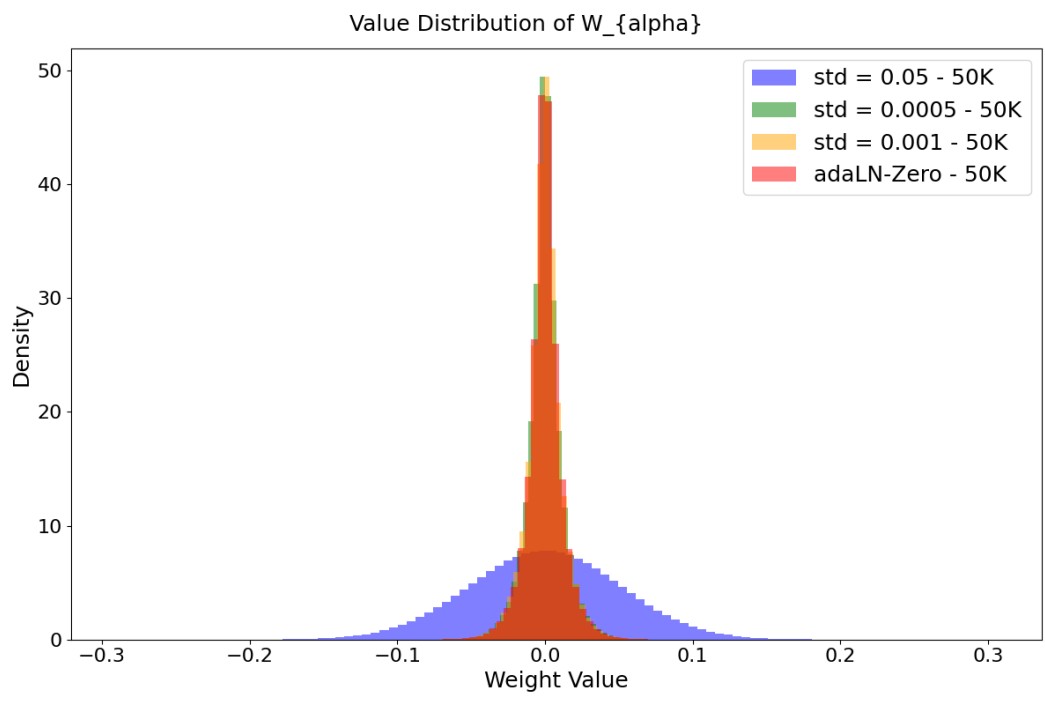

Figure 21: Value distributions of $W_\alpha$ with different std in Gaussian initialization.

## A.10 ADALN-GAUSSIAN-V2

We begin by considering $std(1e-3, 2e-3, 8e-4)$ [10], restrict from 8e-4 to 2e-3 inspired by Tab. 2, and perform a grid search in Tab. 7. It is observed that $std(8e-4, 1.2e-3, 8e-4)$ produces the best FID. We denote this initialization as *adaLN-Gaussian-v2*.

Based on adaLN-Gaussian-v2, we further explore a more sophisticated block-wise initialization. This is motivated by our observation that the peak value and bottom width of $W_\alpha^L, W_\gamma^L$, and $W_\beta^L$ varies across DiT blocks in Fig. 6, Fig. 12, and Fig. 13, indicating that different blocks may prefer different std. At our preliminary attempt in App. A.11, we show that block-wise initialization is inferior to the base setting in FID but outperforms the base setting in IS. This highlights the potential of block-wise initialization and requires more effort which we leave as future work.

Table 7: Results of independent std settings for $W_\alpha, W_\gamma, W_\beta$. 0, 0, 0: adaLN-Zero

| Std ($W_\alpha, W_\gamma, W_\beta$) | FID | IS |
|---|---|---|
| 0, 0, 0 | 78.99 | 14.19 |
| 1e-3, 2e-3, 8e-4 | 78.22 | 14.37 |
| 1e-3, 1.2e-3, 8e-4 | 76.57 | **15.01** |
| 8e-4, 1.2e-3, 8e-4 | **76.12** | 14.90 |
| 8e-4, 1.2e-3, 1e-3 | 77.18 | 14.85 |
| 8e-4, 1e-3, 8e-4 | 80.31 | 14.23 |
| 8e-4, 1.4e-3, 8e-4 | 77.53 | 14.54 |
| 8e-4, 1.6e-3, 8e-4 | 78.24 | 14.55 |
| 8e-4, 1.6e-3, 4e-4 | 79.03 | 14.31 |

Furthermore, we compare the performance of adaLN-Gaussian-v2 with adaLN-Zero and adaLN-Gaussian under longer training time as shown in Tab 8. It is seen that adaLN-Gaussian-v2 also outperforms adaLN-Zero, further verifying the effectiveness of our strategy of Gaussian initialization. On the other hand, considering that adaLN-Gaussian achieves superior results to that of adaLN-Gaussian-v2 and is easier to implement, we primarily use adaLN-Gaussian in Tab 4.

| Model | Initialization | CFG | Steps | FID↓ | sFID↓ | IS↑ | Precision↑ | Recall↑ |
|---|---|---|---|---|---|---|---|---|
| DiT-XL/2 | adaLN-Zero | 1 | 400K | 20.02 | 6.09 | 67.34 | 63.33 | 63.06 |
| DiT-XL/2 | adaLN-Gaussian | 1 | 400K | 17.86 | 6.06 | 73.07 | 64.51 | 62.64 |
| DiT-XL/2 | adaLN-Gaussian-v2 | 1 | 400K | 18.77 | 6.08 | 70.07 | 63.92 | 62.72 |

Table 8: Comparison among adaLN-Zero, adaLN-Gaussian, and adaLN-Gaussian-v2.

## A.11 A PRELIMINARY EXPLORATION OF BLOCK-WISE INITIALIZATION

We dive into every block in DiT and find that there also exist discrepancies in peak value among different $W_\alpha^L$ in Fig. 6. $W_\gamma^L$ and $W_\beta^L$ also hold in Fig. 12 and Fig. 13. Generally, the greater the peak value is, the smaller the std is, motivating us to design a more sophisticated block-wise initialization strategy. Specifically, we record the peak value in all blocks for $W_\alpha^L$, $W_\gamma^L$, and $W_\beta^L$, respectively, and use three heuristic polynomial functions to fit these points as shown in Fig. 22. For $W_\alpha^L$, we use 7th degree polynomial whose coefficients are $[-2.49635921e-6, 1.24680129e-4, 1.17149262e-3, -1.70585560e-1, 3.63484494, -2.94971466e+1, 6.74700382e+1, 4.85897902e+2]$. For $W_\gamma^L$ and $W_\beta^L$, we use 5th degree polynomial. Their coefficients are $[-2.46024908e-4, 2.39970674e-2, -8.67912602e-1, 1.45429227e+1, -1.08645122e+2, 4.11540316e+2]$ and $[-1.21059796e-4, 1.09417700e-2, -3.25623123e-1, 4.15804173, -2.00345083e+1, 4.20334676e+2]$, respectively. For $W_\alpha^L$ in L$-th$ block, we use the following formula to calculate its std value:

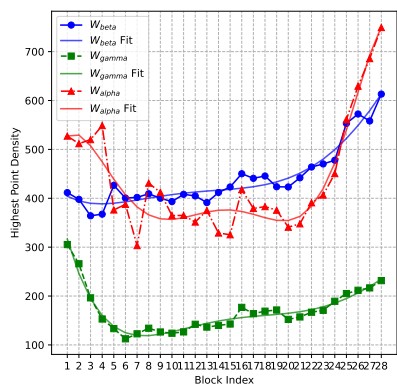

Figure 22: Three polynomial functions to fit the peak values of $W_\alpha$, $W_\gamma$, and $W_\beta$ in all blocks.

---

[10]We empirically find that $N(0, 1e-3)$ closely matches the shape of the $W_\alpha$ distribution in Fig. 4 (b) (10K iterations). Therefore, based on this observation, we begin our further refinement by estimating the std for $W_\gamma$ and $W_\beta$ with their corresponding distribution shapes in 10K iterations.

$$\sigma_\alpha^L = 0.0008/(Poly_\alpha(L)/449.9321) \,, \tag{24}$$

where $Poly_\alpha$ is the polynomial function for $\alpha$, 0.0008 is the base std inherited from Tab. 7, and 449.9321 is the averaged peak value across $W_\alpha^L$ in all blocks.

Similarly, for $W_\gamma^L$ and $W_\beta^L$, we use the following formulas to calculate their std value, respectively:

$$\sigma_\gamma^L = 0.0012/(Poly_\gamma(L)/444.8248) \,, \tag{25}$$

$$\sigma_\beta^L = 0.0008/(Poly_\beta(L)/175.0044) \,. \tag{26}$$

Table 9: Results of block-wise initialization. ✔: with block-wise

| Std $(W_\alpha, W_\gamma, W_\beta)$ | FID | IS |
|---|---|---|
| 8e-4, 1.2e-3, 8e-4 | **76.12** | 14.90 |
| ✘, ✔, ✔ | 79.28 | 14.35 |
| ✔, ✔, ✔ | 76.63 | **14.96** |

We first consider employing block-wise initialization for $W_\gamma^L$ and $W_\beta^L$ since they are well fitted and use 0.0008 for $W_\alpha^L$ by default. Afterward, we initialize them all in a block-wise manner. As shown in Tab. 9, block-wise initialization is inferior to the base setting in FID50K but outperforms the base setting in IS. We leave more exploration as future work.

## A.12 MORE EXPERIMENTS ON EFFECTIVENESS

To further show the effectiveness of adaLN-Gaussian, we add more experiments on other datasets including Tinyimagenet (Le & Yang, 2015), AFHQ (Choi et al., 2020), and CelebA-HQ (Karras et al., 2018) using the best-performing DiT-XL/2 with 50K training steps while keeping all training settings. Moreover, we also use another DiT-based model SiT-XL/2 (Ma et al., 2024b) training on ImageNet1K $256x256$ for 50K to further show the effectiveness and generalization of adaLN-Gaussian. We report all the FID results in Tab. 10. These results show that adaLN-Gaussian consistently outperforms adaLN-Zero, demonstrating the effectiveness of our method.

| | Tiny ImageNet | AFHQ | CelebA-HQ | ImageNet1K (SiT-XL/2) |
|---|---|---|---|---|
| adaLN-Zero | 37.11 | 13.52 | 8.01 | 71.90 |
| adaLN-Gaussian | 36.07 | 12.58 | 7.54 | 67.15 |

Table 10: Comparisons between adaLN-Zero and adaLN-Gaussian on another three datasets including Tinyimagenet, AFHQ and CelebA-HQ, and another DiT-based model SiT. We set CFG=1.

