# OpenReview forum: "Unveiling the Secret of AdaLN-Zero in Diffusion Transformer"
_ICLR.cc/2025/Conference — Submitted to ICLR 2025_

### Official Review · Reviewer_nDDD · 2024-10-23

**Soundness:** 2
**Presentation:** 3
**Contribution:** 1
**Rating:** 3
**Confidence:** 4

**Summary:**

This paper investigates AdaLN-Zero within the DiT architecture and proposes three potential reasons for the performance difference between AdaLN-Zero and AdaLN: the incorporation of an SE-like structure, the use of an effective zero-initialized value, and a gradual weight update order. The authors conduct a series of analyses, ultimately concluding that the zero initialization is the most significant factor. Building on these findings, they introduce a method that leverages Gaussian distributions to initialize each condition modulation, referred to as adaLN-Gaussian. Extensive experiments demonstrate an improvement of 2.16% in FID on ImageNet1K.

**Strengths:**

The presentation is clear, three potential reasons for the performance difference between AdaLN-Zero and AdaLN are studied in detail, and the designed experiment is reasonable.

**Weaknesses:**

1. The contribution of this paper is limited; the only significant advancement is the introduction of adaLN-Gaussian, which leverages Gaussian distributions to initialize each condition modulation in the DiT architecture. The performance gains are also limited.
2. The experiments conducted are insufficient. Training for 800k iterations may not be adequate for the convergence of the DiT model. Given the 2% improvement in the current results at 400k iterations, it raises doubts about whether Gaussian initialization will outperform zero initialization in terms of final performance. Furthermore, to demonstrate the broader applicability of this method, I recommend conducting additional experiments on transformer-based models such as SiT[1] and PixArt-alpha[2] (text-to-image).

[1] Ma, Nanye, et al. "Sit: Exploring flow and diffusion-based generative models with scalable interpolant transformers." arXiv preprint arXiv:2401.08740 (2024).
[2] Chen, Junsong, et al. "Pixart-$\alpha $: Fast training of diffusion transformer for photorealistic text-to-image synthesis." arXiv preprint arXiv:2310.00426 (2023).

**Questions:**

1. How do you summarize the differences between AdaLN-Zero and adaLN as the three points?
2. In Fig. 4, it appears that regardless of the type of initialization used, the weight distribution converges to a similar state after a certain number of training iterations. This raises the question of why the choice of initialization has such a significant impact on performance.
3. A few minor suggestions: 1) Simplify Figure 6 by merging the subgraphs to enhance information density. 2) Rotate Figure 2 to improve the visibility of the text.

---

> ### Author Response · Authors · 2024-11-20
> **Response to Reviewer nDDD (1/2)**
>
> **Q1:** The contribution of this paper is limited; the only significant advancement is the introduction of adaLN-Gaussian
>
> **A1:** **We respectfully disagree with the reviewer's comment.**
>
> Our paper is not merely about a method proposal but, more importantly, an analytical study. Specifically, besides the introduction of adaLN-Gaussian, our detailed analysis about adaLN-Zero also can not be ignored because it unveils the reason why adaLN-Zero outperforms adaLN, which enhances the community's understanding. Moreover, our analysis is the basis of the introduction of adaLN-Gaussian.  It would be inappropriate to neglect the contribution of our analysis.
>
> **In contrast, our analysis is acknowledged by other reviewers.** For example, it is acknowledged by Reviewer [p8jh] that we provide sufficient evidence to support the exploration inside the adaLN-zero. And Reviewer [KCHJ] also comments that our analysis provides some inspiring conclusions.
>
> **Q2:** The performance gains are also limited.
>
> **A2:** **We may not agree with the reviewer's comment about the performance gains of our methods.**
>
> The improvement of our method is acknowledged by reviewer [ZpLc] who comments that our work is simple and effective, achieving remarkable performance.
>
> If the reviewer thinks that our performance gains, e.g., **2.16 FID in 400K**, are limited, how does the reviewer think about the improved performance of SiT[1] accepted by ECCV 2024? For example, SiT-XL/2 outperforms DiT-XL/2 by **2.3 FID in 400K** in Table 1 in its paper (https://arxiv.org/pdf/2401.08740). (Our method and SiT both use the same structure of DiT.)
>
> We sincerely appreciate the reviewer's time and effort, and we have no intention of offending the reviewer. However, to some extent, these comments seem to be unreasonable.
>
> We hope that the reviewer could reevaluate the contributions of our work. As acknowledged by reviewer [ZpLc], our work is simple and effective (with only one line code replaced). Moreover, our method in Table 4 of our paper has shown great generalization to various DiT variants and DiT-based models.
>
> [1] Sit: Exploring flow and diffusion-based generative models with scalable interpolant transformers. ECCV2024
>
> **Q3:** The experiments conducted are insufficient. Training for 800k iterations may not be adequate for the convergence of the DiT model. Given the 2% improvement in the current results at 400k iterations, it raises doubts about whether Gaussian initialization will outperform zero initialization in terms of final performance.
>
> **A3:** We kindly remind the reviewer that in this paper we do not alter the learning algorithm and any structure of DiT model, but only initialize its condition mechanism differently. **Therefore, theoretically, if there is no limit on training steps and excluding local optimum, the finally converged performance of adaLN-Zero would be similar to the performance of adaLN-Gaussian because the model capacity is the same.** However, as reviewer [KCHJ] points out, different initialization methods mainly affect the convergence speed. Under the same training steps, our method allows DiT model to converge faster. For example, adaLN-Gaussian achieves an FID of 14.84 after 600K training steps, while adaLN-Zero requires 800K steps to reach a similar FID of 14.73. When further extending the training time,  adaLN-Gaussian  achieves an FID of 10.83 after 1.5M training steps while adaLN-Zero achieves requires around 2.4M steps to reach a similar FID of 10.67.
>
>
> **Q4:** Conducting additional experiments on transformer-based models such as SiT and PixArt-alpha (text-to-image).
>
> **A4:** We thank the reviewer's suggestion. SiT and PixArt-alpha are both very excellent work in the community. Due to the limited time,  high GPU computing demand (64 V100 & 26 days), and large amount of internal data used in PixArt-alpha, we select to perform additional experiments on transformer-based SiT.  We use the best-performing SiT-XL/2 training on ImageNet1K 256x256 for 50K.  adaLN-Zero produces 71.90 for FID, while adaLN-Gaussian yields 67.15 FID, significantly outperforming adaLN-Zero and demonstrating the effectiveness of our method. Moreover, we also perform experiments using DiT-XL/2 training on other datasets including Tiny ImageNet, AFHQ, and CelebA-HQ for 50K. We report all the results below. These results further show the effectiveness and generalization of adaLN-Gaussian.
>
> | Dataset | Tiny Imagenet|  AFHQ | CelebA-HQ | ImageNet1K (SiT-XL/2)
> |----------|----------|----------|----------| ----------|
> | adaLN-Zero | 37.11 |   13.52 | 8.01 | 71.90 |
> | adaLN-Gaussian | 36.07 |   12.58 |  7.54 | 67.15|

---

> ### Author Response · Authors · 2024-11-20
> **Response to Reviewer nDDD (2/2)**
>
> **Q5:** How do you summarize the differences between AdaLN-Zero and adaLN as the three points?
>
> **A5:** Essentially, our summarized three points reflect the roles of the two additional steps of adaLN-Zero compared to adaLN: 1) introducing scaling element $\alpha$ 2) zero-initializing corresponding linear layers.
>
> The first step changes the structure of DiT model. Through our closer observation of the formed structure, we find that the modified structure shares similarities with the SE module (The first point).
>
> Our last two points are derived from the second step. First of all, we know that initialization has a basic role, namely, determining the initial value of the module weight (The second point).  Particularly, for zero initialization, as previous work[2][3] suggested, it additionally can nullify certain output pathways at the beginning of training, thereby causing the signals to propagate through identity shortcuts. A direct way to reveal the influence of this shortcut behavior on model optimization is to analyze through the lens of gradient update (The third point).
>
> Finally, our step-by-step and decoupling experiments and analysis show that these three points are all related to the improved performance of DiT model.
>
> [2] Accurate, large minibatch sgd: Training imagenet in 1 hour. Arxiv2017
>
> [3] Scalable Diffusion Models with Transformers. ICCV2023
>
> **Q6:** In Fig. 4, regardless of the type of initialization used, the weight distribution converges to a similar state after a certain number of training iterations. Why the choice of initialization has such a significant impact on performance?
>
> **A6:** We think there may be two reasons. Firstly, the results presented in Figure 5 of the DiT paper (https://arxiv.org/pdf/2212.09748), such as the significant performance improvement of adaLN-Zero over adaLN in FID, indicates that the condition mechanism is essential for achieving superior metric outcomes. Hence, leveraging a more appropriate initialization allows condition mechanism to learn better and thereby help DiT model to obtain superior outcomes.
>
> Second, though the weight distribution converges to a similar state after a certain number of training iterations, e.g. 400K in Fig 4 where their distributions are very close to each other, the element-wise discrepancy may be still large. We average the absolute values ​​of the differences between each element in all weight matrices corresponding to adaLN-Zero and adaLN-Mix. The averaged element-level value is 0.05, which indicates that there is still a large element-level value shift in weight matrice. This may be the reason to the performance gap between adaLN-Zero and adaLN-Mix. We also calculate the average difference of the whole model weights between adaLN-Zero and adaLN-Mix. The result is 0.051, indicating there is also a large element-level value shift in the whole model.
>
> **Q7:** A few minor suggestions. 1) Simplify Figure 6 by merging the subgraphs to enhance information density. 2) Rotate Figure 2 to improve the visibility of the text.
>
> **A7:** We thank the reviewer's suggestions.  We follow the reviewer's suggestion to rotate Figure 2, which indeed significantly improves the visibility of the text. As for simplifying Figure 6 by merging the subgraphs, we have tried to do so, e.g., by merging 4 subgraphs. However, while enhancing information density,  the resulting figure is not clear to see the distribution of each block. This may not match the purpose of showing Figure 6 where we aim to demonstrate that $W^{L}_{alpha}$ in each block of DiT exhibits a Gaussian-like distribution.

---

> > ### Comment · Reviewer_nDDD · 2024-11-21
> >
> > I do not neglect the detailed analysis of adaLN-Zero presented in this manuscript, which I have summarized in the Strengths section and considered in my final review. My primary concern is that, as you mentioned, the theoretically converged performance of adaLN-Gaussian proposed in this paper is expected to be similar to that of adaLN-Zero; however, there is no experimental support for this claim. You referenced SiT, but Table 1 in SiT paper shows that SiT's performance improvement of FID at 400k steps is significantly better than yours, and they provide evidence indicating that their final results are superior. The 400k experiments conducted in the current manuscript are insufficient to support your conclusion.
> >
> > If adaLN-Gaussian can demonstrate that it requires fewer iterations to achieve a FID of 2.27, as seen with adaLN-Zero, it would substantiate your claim of reduced convergence time, which would be a meaningful contribution. In my research on DiT, I have observed that early convergence does not always lead to better final results; ultimately, what matters is the performance of the model at convergence, which raises my concerns.
> >
> > In this work, adaLN-Gaussian represents the core contribution. The research on the differences between adaLN-Zero and adaLN is aimed at developing adaLN-Gaussian. Therefore, without a comprehensive demonstration of adaLN-Gaussian, I believe the contribution is quite limited. Parameter initialization for adaLN is indeed an intriguing topic that could provide valuable insights into the rapidly evolving field of image and video generation. Given its importance, I hope you can refine the manuscript before publishing it.
> >
> > I would be happy to continue this discussion with you.

---

> > > ### Author Response · Authors · 2024-11-21
> > > **New Response to Reviewer nDDD**
> > >
> > > We sincerely thank the reviewer for the comment and are very delighted to engage in this discussion!
> > >
> > > We understand the reviewer’s concerns and still continue our training and evaluation, which may require some time. We will report our results promptly during the discussion period as requested by the reviewer.

---

> > > ### Author Response · Authors · 2024-12-02
> > > **Response to Reviewer nDDD about adaLN-Gaussian Convergence Concern**
> > >
> > > Dear Reviewer nDDD,
> > >
> > > We sincerely appreciate your patience and apologize for the delay in responding to your question. Over the past few days, we keep other settings unchanged and continue training our adaLN-Gaussian initialized DiT.
> > >
> > > We find that adaLN-Gaussian achieves 10.09 FID in 2300K training steps, outperforming adaLN-Zero in 2352K training steps (10.67 FID in its paper).
> > >
> > > More importantly, as the reviewer requested, using the same cfg=1.5 as adaLN-Zero, we find that adaLN-Gaussian achieves 2.27 FID in 5400K training steps, less than 7000K in adaLN-Zero. saving around 23% training time with simply one line code replaced.
> > >
> > > We hope our response is not too late to address your concerns and could help you reassess the contributions of our work. Also, we promise to add these results in our final version and would greatly appreciate it if you could consider kindly raising your rating!  Thank you once again for your patience.
> > >
> > > Sincerely,
> > >
> > > Authors

---

### Official Review · Reviewer_KCHJ · 2024-10-30

**Soundness:** 3
**Presentation:** 3
**Contribution:** 3
**Rating:** 6
**Confidence:** 4

**Summary:**

This paper investigates adaLN-Zero, a key component of DiT. They find three key factors that contribute to the superior performance of adaLN-Zero: an SE-like structure, a good zero-initialized value, and a gradual weight update order. The second one plays the most important role. Finally, they propose adaLN-Gaussian, which achieves better results.

**Strengths:**

1. This paper is easy to follow, and the experiments are relatively sufficient.
2. This paper provides some inspiring conclusions.

**Weaknesses:**

1. Part of the description needs further explanation, e.g. line 208：They also zeros out weights of all modulations including Wγ1,Wβ1, Wγ2, and Wβ2 in a block, rendering γ1, β1, γ2, and β2 zero.
2. Some of the inferences are intuitive, and it would be better if there were more rigorous analysis. e.g. line 448：Moreover, this moment should be neither too late, ... ,nor too early, as there may be minimal difference from zero-initialization.

**Questions:**

1. From line 129~130, we know that all alpha are initialized to 0. Can you expalin line 209: why are gama1, beta1, gama2, beta2 also zero?
2. In Table 4, if there is no limit on steps, will the results of adaLN-Zero catch up with adaLN-Gaussian when training for more steps? Do you have any relevant experimental results? Maybe different initialization methods mainly affect the convergence speed.
3. In addition to adaLN-Zero, have you analyzed the distribution of parameters of other parts of DiT? Do they also end up being Gaussian? How were these parameters initialized?

---

> ### Author Response · Authors · 2024-11-20
> **Response to Reviewer KCHJ (1/2)**
>
> **Q1:** From line 129~130, we know that all alpha are initialized to 0. Can you expalin line 209: why are gama1, beta1, gama2, beta2 also zero?
>
> **A1:** Why are gama1, beta1, gama2, beta2 also zero? The answer is that in adaLN and adaLN-Zero, as shown in Fig 2 of our paper, the weights of the linear layer that produces gama1, beta1, gama2, and beta2 are initialized to zero at the beginning. Sorry for the confusion. We have revised the corresponding part of the paper to make it clearer.
>
> **Q2:** Some of the inferences are intuitive, and it would be better if there were more rigorous analysis. e.g. line 448：Moreover, this moment should be neither too late, ... ,nor too early, as there may be minimal difference from zero-initialization.
>
> **A2:** We thank the reviewer for the constructive suggestions. To make the inference more rigorous,  we analyze two representative std settings including $std=0.05$ and $std=0.0005$, which correspond to a late moment (a large std) and an early moment (a small std), respectively.
>
> We first illustrate their weight distributions of $W_{\alpha}$ in the conditioning mechanism and compare them with that of adaLN-Zero and adaLN-Gaussian ($std=0.001$). The results are shown in **Fig. 21** of our revised paper in Appendix.9.  It can be seen that a large std $std=0.05$ presents a relatively uncompact distribution and exhibits a significant discrepancy in distribution shape compared to the rest settings. This result indicates that a large std may be incompatible with other parameters, resulting in a slow speed of convergence and a poor performance. Moreover, we consider this a step further. Theoretically, if we further increase the std value, it would become close to the default initialization in adaLN-Step1 (xavier_uniform) while the performance of adaLN-Step1 is also bad.
>
> For a small std $std=0.0005$, it can be seen that the distribution of $W_{\alpha}$ is quite similar to that of adaLN-Zero and adaLN-Gaussian ($std=0.001$). However, there still exists a slight discrepancy. To make this discrepancy clearer, we average the absolute values ​​of the differences between each element in $W_{\alpha}$ corresponding to $std=0.0005$ and adaLN-Zero, and $std=0.0005$ and adaLN-Gaussian. The element-wise averaged results are 0.0121 and 0.0124, respectively. By comparing the results (0.0121 < 0.0124), it is shown that small std leads to weights relatively closer to that of zero-initialization (adaLN-Zero). And, to some extent, the corresponding performance also proves it where $std = 0.0005$ produces  80.68 for FID, closer to adaLN-Zero (78.99) compared to adaLN-Gaussian (76.21).
>
> We thank the reviewer again and have revised the corresponding part in our paper (as highlighted in blue), which makes our paper clearer and reasonable.
>
> **Q3:** If there is no limit on steps, will the results of adaLN-Zero catch up with adaLN-Gaussian when training for more steps? Do you have any relevant experimental results? Maybe different initialization methods mainly affect the convergence speed.
>
> **A3:**  Yes, if there is no limit on steps, the performance of adaLN-Zero could catch up with that of adaLN-Gaussian when training for more steps. For example, adaLN-Zero training for 800K produces 14.73 FID, catching up with adaLN-Gaussian training for 600K (14.84 FID).
>
> And we agree with the reviewer's comment that different initialization methods mainly affect the convergence speed. Essentially, adaLN-Gaussian only changes the initialization strategy and does not alter any structure of DiT model, which means the model's capacity is the same. Therefore, theoretically, adaLN-Gaussian will not significantly influence the final converged performance but converge faster.  For example, adaLN-Gaussian achieves an FID of 14.84 after 600K training steps, while adaLN-Zero requires 800K steps to reach a similar FID of 14.73. When further extending the training time,  adaLN-Gaussian  achieves an FID of 10.83 after 1.5M training steps while adaLN-Zero achieves requires around 2.4M steps to reach a similar FID of 10.67.

---

> ### Author Response · Authors · 2024-11-20
> **Response to Reviewer KCHJ (2/2)**
>
> **Q4:** Have you analyzed the distribution of parameters of other parts of DiT? Do they also end up being Gaussian? How were these parameters initialized?
>
> **A4:** Yes, we have analyzed the weight distribution of other parts of DiT including Attention and Mlp in DiT Block, PatchEmbed, LabelEmbedder, and TimestepEmbedder in Appendix A.7. In our experiments, we find that all of their weight distributions except PatchEmbed end up being Gaussian-like.
>
> For the last question (How were these parameters initialized?) In DiT code, Attention and Mlp in DiT Block, and PatchEmbed are with Xavier uniform. LabelEmbedder and TimestepEmbedder are initialized with normal distribution.
>
> Naturally, we can consider Gaussian initializations for these modules as well except PatchEmbed to accelerate training. For example, we could uniformly use Gaussian initialization for Attention and Mlp in DiT Block. We set the mean to 0 and use several choices for std such as 0.001, 0.01, 0.02, 0.03, and 0.04. We use DiT-XL-2 and train for 50K steps for simplicity. The results are shown below.
>
> | Std | Default| 0.001 | 0.01 | 0.02 | 0.03 | 0.04 |
> |----------|----------|----------|----------| ----------|----------| ----------|
> | FID | 76.21 | 92.09 |  85.28 | 80.89 | 91.21 | 98.50 |
>
> We see that the performance is inferior to the default initialization. Therefore, more precise hyperparameter tuning may be needed for these modules to further improve the performance, which we leave as future work.
>
> We hope our response will clarify the reviewer's confusion and alleviate the concern. And we sincerely hope to obtain support from the reviewer.

---

> ### Author Response · Authors · 2024-12-02
> **A Kind Reminder to Reviewer KCHJ about our rebuttal**
>
> Dear Reviewer KCHJ,
>
> We hope that this letter finds you well.
>
> As the end of this discussion is going to be close, we would greatly appreciate it if you could let us know whether our responses and revisions align with your expectations and have addressed your concerns.  We may need to highlight our new experiment where adaLN-Gaussian achieves 2.27 FID in 5400K training steps, less than 7000K in adaLN-Zero which is the longest training steps in DiT paper. This result demonstrates that our method adaLN-Gaussian is also superior over adaLN-Zero in the scaled training for final convergence.
>
> If there are any remaining issues where further clarification is needed, please don’t hesitate to let us know. We are more than willing to provide additional explanations.
>
> Thank you once again for your time and effort. We look forward to hearing your thoughts and would greatly appreciate it if you could consider kindly raising your rating.
>
> Sincerely,
>
> Authors

---

### Official Review · Reviewer_p8jh · 2024-10-31

**Soundness:** 4
**Presentation:** 3
**Contribution:** 2
**Rating:** 6
**Confidence:** 4

**Summary:**

This work focuses on an overlooked part in the diffusion transformer - the zero initialization of adaptive LayerNorm.
Authors start from the similarity between SENet with DiT's adaLN, and develop some variants from adaLN.
After several discussions about the gradient update over the weight of adaLN and the summarization of the benefits from adaLN-zero, authors  propose to leverage Gaussian distributions to initialize the adaLN.
The exps over IN-1K and DiT-based backbones are sufficiently perferformed.

**Strengths:**

1. The motivation is very good (the overlooked part in DiT) and the issue in this work in quite interesting.

2. Clear clarity and good presentation.

3. Provide sifficient evidence to support the exploration inside the adaLN-zero, not only the gradient update but also the varients of adaLN.

4. Good and fair experimentss to support the proposed method. Authors provide relatively big scale exps on IN-1K 512X512 which is expensive, and the backbones are mainly based on large scale DiT. Besides authors also evaluate the effectiveness on other DiT-based backbones.

**Weaknesses:**

1. The start point is very unclear for me. How can you find the similarity between adaLN and SE archtecture？
I consider that this start point should be better explained and provide a structure figure of SE archtecture.

**Questions:**

1. How can you find the similarity between adaLN and SE archtecture？ This is an intersting point. Hope that authors can provide some principles but not intuitions.

2. As shown in Figure2, the performance of Gaussian initialization is a U-shaped trending. Could you please some analysis about this trending? Why the large std can brings a relatively bad results.

---

> ### Author Response · Authors · 2024-11-20
> **Response to Reviewer p8jh**
>
> **Q1:** How can you find the similarity between adaLN and SE archtecture？ This is an intersting point. Hope that authors can provide some principles but not intuitions.
>
> **A1:** We thank the reviewer's comment and we are very willing to explain how we find such similarity. We primarily find the connections between them from three aspects: 1) overall structure; 2) module function; and 3): detailed mathematical formula.
>
> First,  from the view of the overall structure, the adaLN-Zero, and SE module both serve as a side pathway compared to the main path.
>
> Then, from the view of the module function, scaling element $alpha$ in adaLN-Zero plays a similar role as the SE module, both of which aim to perform a channel-wise modulation operation.  Hence, to achieve it, they may yield outputs of the same structure, e.g., vector,  thereby sharing some similarity in output formulation.
>
> Finally, from the view of the mathematical formula, though we may not directly find the similarity due to the existence of the SiLU function, a more detailed expandation allows us to find a close connection in the mathematical formula between the adaLN-Zero and SE module. With a slight adjustment according to experience, the similarity appears.
>
> We thank the reviewer's question and have added the response as well as a figure of the structure of the SE module to our paper (as highlighted in blue), which makes the start point clearer.
>
> **Q2:** As shown in Figure2, the performance of Gaussian initialization is a U-shaped trending. Could you please some analysis about this trending? Why the large std can brings a relatively bad results.
>
> **A2:** The reviewer may want to say Table 2 instead of Figure 2. Intuitively, since the weights of the conditional mechanisms we counted are Gaussian-like distributions, there should exist an optimal std hyperparameter when initializing these weights with Gaussian, and naturally, the values ​​on both sides of this hyperparameter are relatively unsuitable.
>
> We follow the reviewer's advice and analyze this U-shaped trending by leveraging two representative settings, i.e., $std=0.0005$ and $std=0.05$, which are the two ends of U-shaped results.
>
> We first illustrate their weight distributions of $W_{\alpha}$ in the conditioning mechanism and compare them with that of adaLN-Zero and adaLN-Gaussian ($std=0.001$). The results are shown in **Fig. 21** of our revised paper in Appendix.9. We find that a large std $std=0.05$ presents a relatively uncompact distribution and exhibits a significant discrepancy in distribution shape compared to the rest settings. This result indicates that a large std may increase the difficulty of optimization, resulting in a slow speed of convergence. Moreover, we consider this a step further. Theoretically, if we further increase the std value, it would become close to the default initialization in adaLN-Step1 (xavier_uniform) while the performance of adaLN-Step1 is also bad.
>
> For a small std $std=0.0005$, we find that the distribution of $W_{\alpha}$ is more compact and is quite similar to that of adaLN-Zero and adaLN-Gaussian ($std=0.001$). Though similar, there still exists a slight discrepancy. To make this discrepancy clearer, we average the absolute values ​​of the differences between each element in $W_{\alpha}$ corresponding to $std=0.0005$ and adaLN-Gaussian, and adaLN-Zero and adaLN-Gaussian. The results are 0.0124 and 0.0122, respectively, which indicates that there is still an element-level value shift in weight matrice. The reason that the performance of $std=0.0005$ is relatively bad in Table 2 may possibly be because $W_{\alpha}$ in $std=0.0005$ is farther away from that in adaLN-Gaussian compared to adaLN-Zero since 0.0124>0.0122. Note that though the difference is small, considering the large number of elements (about 74.3M), the influence may not be ignored.
>
> We hope our response will clarify the reviewer's confusion. And we sincerely hope to obtain support from the reviewer.

---

> > ### Comment · Reviewer_p8jh · 2024-11-22
> >
> > Thanks for you detailed answers!  My questions have been addressed.
> > And I consider such an interesting work should be appeared in ICLR 2025.
> > However, I cannot judge whether the proposed gaussian initializing is a trick or can be popularily applied in practical DiT-based applications and scaled training.
> > So I keep the ratings (6).

---

> > > ### Author Response · Authors · 2024-12-02
> > > **New Response to Reviewer p8jh about the scaled training**
> > >
> > > Dear Reviewer p8jh
> > >
> > > We thank you for your reply and we also feel sorry for the confusion you mentioned. To better help you evaluate our method, we further extend the training steps to illustrate the efficacy of our method in scaled training.
> > >
> > > Our experiments show that adaLN-Gaussian produces 10.09 for FID in 2300K training steps, outperforming adaLN-Zero in 2352K training steps (10.67 FID in its paper).  Additionally, we find that using the same cfg=1.5 as adaLN-Zero, adaLN-Gaussian achieves 2.27 FID in 5400K training steps, less than 7000K in adaLN-Zero which is the longest training steps in DiT paper. These results show the superiority of adaLN-Gaussian to adaLN-Zero in the scaled training.
> > >
> > > We hope our additional results could remove your concerns and help you to evaluate our method. And we also sincerely hope to obtain support from you during the reviewer/AC discussion.
> > >
> > > Authors

---

### Official Review · Reviewer_ZpLc · 2024-11-04

**Soundness:** 2
**Presentation:** 1
**Contribution:** 3
**Rating:** 5
**Confidence:** 4

**Summary:**

This paper investigates three mechanisms of adaLN-Zero, including the SE-like network structure, zero-initialization, and the weight update order. Based on the analysis of adaLN-Zero, this work proposes an improved initialization method, adaLN-Gaussian, which utilizes Gaussian distributions to initialize the weights of each condition modulation.

**Strengths:**

The proposed adaLN-Gaussian initialization is simple and effective, achieving remarkable performance.

**Weaknesses:**

1. The overall logic of the paper is somewhat disorganized, and there is a logical gap between the analysis of adaLN-Zero and adaLN-Gaussian. Could the authors provide a more detailed explanation of why Gaussian distributions are used to initialize weights?
2. This work lacks mathematical analysis; all conclusions are drawn from experimental results and statistics, and the authors only conducted experiments using the ImageNet dataset. I think more extensive and general experiments are needed to validate the effectiveness of adaLN-Gaussian.

**Questions:**

See Weaknesses.

---

> ### Author Response · Authors · 2024-11-20
> **Response to Reviewer ZpLc**
>
> **Q1:** Could the authors provide a more detailed explanation of why Gaussian distributions are used to initialize weights?
>
> **A1:** Yes, we feel very sorry for the confusion and are very willing to explain the reason for using Gaussian distributions. The logic is that:
>
> First of all, by comparing the differences between adaLN-Zero and adaLN, our analysis studies three elements that collectively drive the performance enhancement: 1) an SE-like structure, 2) zero-initialized value, and 3) a “gradual” update order. Though previous work thinks the manner of shortcuts plays a major role, our analysis finds that it is a good zero-initialized location itself that plays a significant role, which indicates that it is important to find a suitable initialization.
>
> Further, we find that, from the perspective of weight distribution, though weights in the conditioning mechanism are zero-initialized, after a certain number of training steps, they transition from zero distributions to Gaussian-like distributions. Hence, inspired by this, our insight is that we can expedite this distribution shift by directly initializing the weights via a suitable Gaussian distribution to potentially accelerate training.
>
> We have revised the corresponding part of the paper (as highlighted in blue) to make it clearer.
>
> **Q2:** More extensive and general experiments
>
> **A2:** We thank the reviewer for the kind and constructive suggestions. To further show the effectiveness of adaLN-Gaussian, we add more experiments on other datasets including Tinyimagenet, AFHQ, and CelebA-HQ using the best-performing DiT-XL/2 with 50K training steps while keeping all training settings. Moreover, we also use another DiT-based model SiT-XL/2 [1] training on ImageNet1K 256x256 for 50K to further show the effectiveness and generalization of adaLN-Gaussian. We report all the results below. These results show that adaLN-Gaussian consistently outperforms adaLN-Zero, demonstrating the effectiveness of our method.
>
> | Dataset | Tiny Imagenet|  AFHQ | CelebA-HQ | ImageNet1K (SiT-XL/2)
> |----------|----------|----------|----------| ----------|
> | adaLN-Zero | 37.11 |   13.52 | 8.01 | 71.90 |
> | adaLN-Gaussian | 36.07 |   12.58 |  7.54 | 67.15|
>
> We hope our response will clarify the reviewer's confusion and alleviate the concern.  And we sincerely hope to obtain support from the reviewer.
>
> [1] Sit: Exploring flow and diffusion-based generative models with scalable interpolant transformers. ECCV2024

---

> > ### Comment · Reviewer_ZpLc · 2024-11-29
> >
> > Thank you for your response. However, the theoretical analysis I requested is still missing, and I remain unclear about the necessity of using Gaussian distribution for weight initialization. Additionally, in the provided extended experiments, training for only 50K steps is insufficient to demonstrate the effectiveness of adaLN-Gaussian. Can its advantages be maintained over longer training periods?
> >
> > Due to the lack of theoretical analysis and incomplete experiments, I am adjusting my rating to 5. I would be happy to continue the discussion if the authors provide more evidence and experiments.

---

> > > ### Author Response · Authors · 2024-12-02
> > > **Responses to Reviewer ZpLc about theoretical analysis and more experiments (1/2)**
> > >
> > > Dear Reviewer  ZpLc,
> > >
> > > We sincerely appreciate your patience and apologize for the delay in responding to your question.
> > >
> > > Inspired by previous study[1], we consider to investigate the variance of the response of a scale element $\alpha$ in a condition modulation under different initialization strategies including zero-initialization (adaLN-Zero), gaussian initialization (adaLN-Gaussian), and the default xavier uniform initialization (adaLN-Mix).
> > >
> > > For simplicity, we omit the bias term.  Thus, for a scale elemen $\alpha$ as shown in Eq 1 in our paper,  it is formulated as:
> > >
> > > $\alpha = \operatorname{SiLU}(c) * W_{\alpha} = (c \cdot \operatorname{Sigmoid}(c)) * W_{\alpha} $
> > >
> > > $∗$ is matrix multiplication and $\cdot$ is Hadamard product.
> > >
> > > We let the initialized elements in $W_{\alpha}$ be mutually independent and share the same distribution. Similar to [1], we assume that the elements in $c$ are also mutually independent and share the same distribution, and $c$ and $W_{\alpha}$ are independent of each other. Then we have
> > >
> > > $\operatorname{Var}\left[\alpha_{i}\right] = n\operatorname{Var}\left[(c_{i} \cdot \operatorname{Sigmoid}(c_{i})) * w_{\alpha} \right] $
> > >
> > > where $n$ is a scalar that represents number of neural connections,  $\alpha_{i}$, $c_{i}$, and $w_{\alpha}$ represent the random variables of each element in $\alpha$, $c$, and $W_{\alpha}$, respectively. We know that all initialization strategies allow $w_{\alpha}$ to have zero mean.  Then the variance of the product of independent variables gives us:
> > >
> > > $\operatorname{Var}\left[\alpha_{i}\right] = n \operatorname{Var}\left[w_{\alpha} \right] \operatorname{E}\left[(c_{i} \cdot \operatorname{Sigmoid}(c_{i}))^{2} \right] $
> > >
> > > Therefore, we can see that the variance of an element in $\alpha$ depends on $\operatorname{E}\left[(c_{i} \cdot \operatorname{Sigmoid}(c_{i})) \right]$ and $n\operatorname{Var}\left[w_{\alpha} \right]$ while $n\operatorname{Var}\left[w_{\alpha} \right]$ is highly related to our method.
> > >
> > > In adaLN-Zero case, $n\operatorname{Var}\left[w_{\alpha} \right]$ could be 0. In adaLN-Gaussian case, $n\operatorname{Var}\left[w_{\alpha} \right]$ is around 0.012. In adaLN-Mix case, $n\operatorname{Var}\left[w_{\alpha} \right]$ is 1.04. We know that at the early stage of training, due to random initialization, $c$ would vary a lot, intoducing a large variance. To some extent, this conditioning input may be unreliable and disturb the model optmization. AdaLN-Zero and adaLN-Gaussian  depress this variance, allowing the model to learn more stably, while adaLN-Mix fails to do so. As a result, AdaLN-Zero and adaLN-Gaussian learn faster than adaLN-Mix.
> > >
> > > [1] X. Glorot and Y. Bengio. Understanding the difficulty of training deep feedforward neural networks. In International Conference on Artificial Intelligence and Statistics, pages 249–256, 2010.

---

> ### Author Response · Authors · 2024-11-29
> **New Response to Reviewer ZpLc**
>
> Dear Reviewer ZpLc:
>
> Thank you for your comment; we are delighted to engage in this discussion! We apologize for the absence of a theoretical analysis. Our experiments are still ongoing and require additional time. We will share the results and provide further evidence promptly within the discussion period.

---

> ### Author Response · Authors · 2024-12-02
> **Responses to Reviewer ZpLc about theoretical analysis and more experiments (2/2)**
>
> Further, to demonstrate that adaLN-Zero indeed forms a Gaussian-like distribution, we employ KL-Divergence to measure the distance between its distribution and a true Gaussian. Specifically, we use the weights of adaLN-Zero at 50K steps to compute its mean and standard deviation. These parameters are then used to initialize a Gaussian distribution, from which we sample the same number of points as adaLN-Zero. Finally, the KL distance between the two sets of sampled points is calculated using the nearest neighbor nonparametric estimation method, as detailed below.
>
> $D_{\text{KL}}(P \| Q) \approx \frac{1}{n} \sum_{i=1}^{n} \log \frac{\rho_i}{\nu_i} + \log \frac{m}{n-1}$
>
> $m$ and $n$ are the number of sample points. $\rho_i$ represents the nearest neighbor distance of point $x_{i}$ in $P$.  $\nu_i$ is similar.
>
> The calculated KL-Div is 0.065. Typically, the closer the two distributions are, the smaller the KL-Div is. If the two distributions are the same, the KL-Div is 0. Therefore, the calculated result demonstrates that adaLN-Zero indeed formulates Gaussian-like distribution.
>
> For the confusion of the necessity of using Gaussian distribution for weight initialization, we feel sorry for this. We think there may be two reasons for this necessity. Firstly, the results in Figure 5 of the DiT paper (https://arxiv.org/pdf/2212.09748), e.g., the significant improvement of adaLN-Zero over adaLN in FID, indicate that the condition mechanism is essential for achieving superior metric outcomes. Therefore, selecting a suitable initialization for the condition mechanism could be necessary. Simultaneously, this is also an easily achieved and low-cost method to help with training. Secondly using Gaussian is motivated by our statistics results: we find that weights in the conditioning mechanism, though zero-initialized, always transition to a Gaussian-like distribution. Therefore, using a suitable Gaussian initialization could better expedite this distribution shift.
>
> Further, we follow your advice to extend our training steps. However,  we may need to explain that in fact 50K iterations (batch size=256) is relatively suitable for Tiny Imagenet, AFHQ, and CelebA-HQ  because the number of their images is small,  only 100K, 14K, and 30K, respectively, compared to ImageNet1K. We feel sorry that we fail to update these information timely and sorry for the confusion this training step setting brings to you.
>
> But we are willing to further extend the training steps. We extend an additional 50K for Tiny Imagenet and 150K for ImageNet1K (SiT-XL/2).  The results are presented below where our method adaLN-Gaussian still outperforms adaLN-Zero when the training step is longer.
>
> | Dataset | Tiny Imagenet |  ImageNet1K (SiT-XL/2) |
> |----------|----------|----------|
> | adaLN-Zero | 32.45 |   31.07 |
> | adaLN-Gaussian | 31.64 |  27.98 |
>
> We do not extend more training steps for Tiny Imagenet, AFHQ, and CelebA-HQ as our experiments show this will lead to overfitting, an increase in FID. For example, an additional 2K training steps for CelebA-HQ increases the FID from 7.54 to 11.20.
>
> In addition, we also extend the training steps of DiT to further show the effectiveness of our method under longer training time. The results are shown below. We find that adaLN-Gaussian produces 10.09 for FID in 2300K training steps, outperforming adaLN-Zero in 2352K training steps (10.67 FID in its paper). Moreover, using the same cfg=1.5 as adaLN-Zero, adaLN-Gaussian achieves 2.27 FID in 5400K training steps, less than 7000K in adaLN-Zero which is the longest training steps in DiT paper. These results further demonstrate the superiority of adaLN-Gaussian over adaLN-Zero.
>
> | Dataset | ImagnetNet (cfg=1)  |  ImagnetNet (cfg=1.5) |
> |----------|----------|----------|
> | adaLN-Zero | 10.67 (2352K) | 2.27 (7000K)   |
> | adaLN-Gaussian | 10.09 (2300K) |  2.27 (5400K) |
>
> We hope our response above could address your concerns and help you reassess our work. We would greatly appreciate it if you could consider kindly raising your rating! Thank you once again for your patience.

---

### Author Response · Authors · 2024-11-20
**General Response**

We sincerely appreciate all reviewers’ time and efforts in reviewing our paper. We are glad to find that reviewers recognized our contributions:

- **Motivation.** Very good motivation and quite interesting work [p8jh]; Easy to follow [p8jh]; Clear presentation[nDDD]
- **Method.** Simple and effective[ZpLc]; Remarkable performance[ZpLc]
- **Experiments and analysis.** Good, fair, reasonable, and sufficient experiments [p8jh, KCHJ, nDDD]; Inspiring conclusions[KCHJ]; Detailed analysis [nDDD]

And we also thank all reviewers for their insightful and constructive suggestions, which helps a lot in further improving our paper.

In addition to the pointwise responses below, we summarize some supporting experiments and analysis added in the rebuttal according to reviewers’ suggestions.

### New experiments and analysis:

- Adding more experiments on other datasets including Tinyimagenet, AFHQ, and CelebA-HQ, and on another model SiT[1]  to show the effectiveness of our method
- Providing more analysis such as about a U-shaped  trending for Gaussian initialization in Table 2
- Using Gaussian initializations for other module parts of DiT
- Longer training steps for DiT-XL/2, e.g., 5.4M, to achieve 2.27, faster than adaLN-Zero which uses 7M steps, the longest steps in DiT paper

The additional experiments and modifications to the language have been delivered in our paper and reflected in the revised version. We hope our pointwise responses below could clarify all reviewers’ confusion and alleviate all concerns. We thank all reviewers’ time again.

[1] Sit: Exploring flow and diffusion-based generative models with scalable interpolant transformers. ECCV2024

---

### Author Response · Authors · 2024-12-04
**A concise summary to assist the AC in evaluating our work**

## The contributions of this work

This work starts from an important but often overlooked module in DiT called adaLN-Zero. Initially introduced in the DiT paper, adaLN-Zero demonstrated superior performance compared to its counterpart, adaLN, which piqued our interest and prompted us to investigate further. Through step-by-step analysis, we provide some interesting conclusions and propose a simple yet effective refinement with one line code replaced, showing a promising pathway for future generative models. Our contributions can be summarized as follows:

- We study three key factors that collectively contribute to the superior performance of adaLN-Zero: an SE-like structure, a good zero-initialized value, and a gradual weight update order. Among them, we find that the a good zero-initialized value plays the most pivotal role.
- Based on the distribution variation of condition modulation weights, we heuristically leverage Gaussian distributions to initialize each condition modulation, termed **adaLN-Gaussian**.
- We conduct comprehensive experiments across various settings to demonstrate adaLN-Gaussian’s effectiveness and generalization including 1) different datasets (ImageNet1K, Tiny Imagnet, AFHQ, and Celeb-HQ),  2) DiT variants (DiT-B, DiT-L, and DiT-XL), and 3) DiT based models (VisionLlama, U-DiT, and SiT). We also involve 4) different training durations. Particularly, our method takes 5400K training steps to achieve 2.27 FID, faster than adaLN-Zero which uses 7000K steps, the longest steps in DiT paper.

## The concerns from reviewers in discussion and how we alleviate them

During the past discussions, the reviewers raised several concerns regarding our paper and our responses. We summarize them and show how we alleviate it as follows:

### Concerns from Reviewer ZpLc

**Concern 1: Lack of Mathematical Analysis**

* The reviewer pointed out that the mathematical analysis they requested was missing.
* **How we alleviate it** :  We provide the mathematical analysis from two aspects.  1) Inspired by previous study[1], we investigate the variance of the response of the scale element  $\alpha$ in a condition modulation under different initialization strategies including zero-initialization (adaLN-Zero), gaussian initialization (adaLN-Gaussian), and the default xavier uniform initialization (adaLN-Mix),  analyzing the relationships between the variance and the performance of different initialization strategies. 2) We use the nearest neighbor nonparametric estimation method to demonstrate that adaLN-Zero indeed forms a Gaussian-like distribution.

  [1] X. Glorot and Y. Bengio. Understanding the difficulty of training deep feedforward neural networks.

**Concern 2: Unclear Necessity of Gaussian Distribution for Weight Initialization**

* The reviewer was unclear about the necessity of using Gaussian distribution for weight initialization.
* **How we alleviate it** : We explain two reasons for the necessity of using Gaussian distribution for weight initialization. First, the significant improvement of adaLN-Zero over adaLN shows the importance of condition mechanism for superior performance. Thus it is necessary to select a more suitable initialization. Secondly using Gaussian is motivated by our statistics results: we find that weights in the conditioning mechanism, though zero-initialized, always transition to a Gaussian-like distribution. Thus, using a suitable Gaussian initialization could better expedite this distribution shift.

**Concern 3: Insufficient Extended Experiments**

* The reviewer noted that the extended experiments with only 50K training steps were insufficient to demonstrate the effectiveness of adaLΝ - Gaussian over longer training periods.
* **How we alleviate it** : We present the results of extended training on different datasets to prove the long - term effectiveness of adaLΝ-Gaussian, e.g., an additional 50K for Tiny ImageNet, 150 K for ImageNet1K  (SiT-XL/2). We do not extend more training steps for Tiny Imagenet, AFHQ, and CelebA-HQ as our experiments show this will lead to overfitting, an increase in FID. We also show that our method takes 5400K to achieve 2.27 FID, faster than adaLN-Zero which uses 7000K steps, the longest steps in DiT paper.

### Concerns from Reviewer p8jh and nDDD (regarding scaled training)

They are both concerned about the scaled training of our method for DiT, namely whether our method could  outperform adaLN-Zero after a long training time and could be faster to achieve the finally converged performance (2.27 FID in DiT).

* **How we alleviate it** :  We report the FID results below which show the superiority of adaLN-Gaussian in the scaled training and it is faster than adaLN-Zero to achieve 2.27 FID, which uses 7000K steps, the longest steps in DiT paper.

| Dataset | ImagetNet (cfg=1)  |  ImagetNet (cfg=1.5) |
|----------|----------|----------|
| adaLN-Zero | 10.67 (2352K) | 2.27 (7000K)   |
| adaLN-Gaussian | 10.09 (2300K) |  2.27 (5400K) |

---

### Meta-Review · Area_Chair_wdRv · 2024-12-16

**Metareview:**

This paper analyzes the DiT architecture of image generation, and in particular the adaLN-Zero conditioning. It identies the zero initialization as critical for performance, and proposes an alternative Gaussian initialization, which is found to stabilize and improve training.
Reviewers appreciate the presentation, and improved performance due to adaLN-Gaussian,
There main concerns relate to the scope of the contribution, analysis and experiments.

**Additional Comments On Reviewer Discussion:**

The authors provided an extensive rebuttal and updates to the manuscript. The reviewers considered the rebuttal which addressed their concerns only partially, and the final recommendations were split between two marginal accept recommendations, and two (marginal) negative recommendations. Beyond the analysis of DiT initialization, the main contribution is empirical exploration of the impact of Gaussian weight initialization, which is observed to accelerate convergence although the paper does not extensively analyze the convergence speed across different initialization methods relative to the final point of convergence.

---

### Decision · Program_Chairs · 2025-01-22

Reject